# PEAR: Pixel-aligned Expressive humAn mesh Recovery

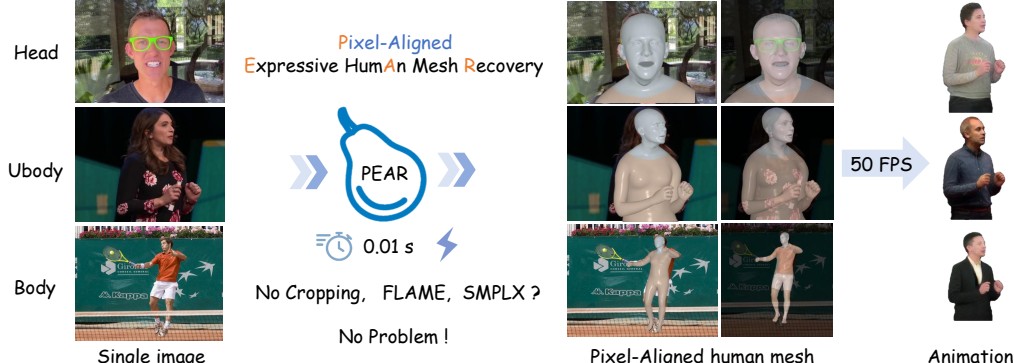

Figure 1: We propose **PEAR**, which achieves pixel-level alignment over previous human mesh recovery methods and demonstrates stronger generalization. It captures more accurate facial details, adapts to diverse inputs, and performs pose recovery within 0.01s from a single image without body-part cropping, providing significant convenience for downstream real-time animation tasks.

## ABSTRACT

Reconstructing 3D human meshes from a single in-the-wild image remains a fundamental challenge in computer vision. Existing methods often produce coarse body poses and exhibit misalignments and unnatural artifacts in fine-grained regions such as the face and hands, which can progressively accumulate and lead to significant errors in downstream tasks. To address this issue, we propose PEAR—a unified framework for human mesh recovery and rendering. PEAR explicitly tackles two major limitations of current methods: inaccurate localization of fine-grained human pose details and insufficient photometric supervision for self-reconstruction. Specifically, we train a Transformer-based model that can recover expressive 3D human geometry (SMPLX + FLAME) from a single image without cropping specific body parts. This preprocessing-free design enables real-time inference at over 100 FPS. Furthermore, we integrate the model with a neural renderer to jointly optimize geometry and appearance, which significantly enhances the reconstruction accuracy of fine-grained human geometry and yields higher-quality rendering results. Lastly, we curate a large-scale dataset of images and videos with human pose and keypoint annotations to facilitate model training. Extensive experiments on multiple benchmark datasets demonstrate that the proposed approach achieves significant improvements in both geometric reconstruction accuracy and rendering quality.

## 1 INTRODUCTION

3D human pose estimation has been a long-standing research focus in computer vision, with wide-ranging applications in robotic perception and interaction Fu et al. (2024); Li et al. (2024), immersive gaming, and virtual human generation for film and live streaming. Recent advances in this field have been largely driven by the introduction of parameterized human body models, notably SMPL Loper et al. (2015), SMPLX Pavlakos et al. (2019a), and GHUM Xu et al. (2020). These models offer compact representations that map high-dimensional human geometry into

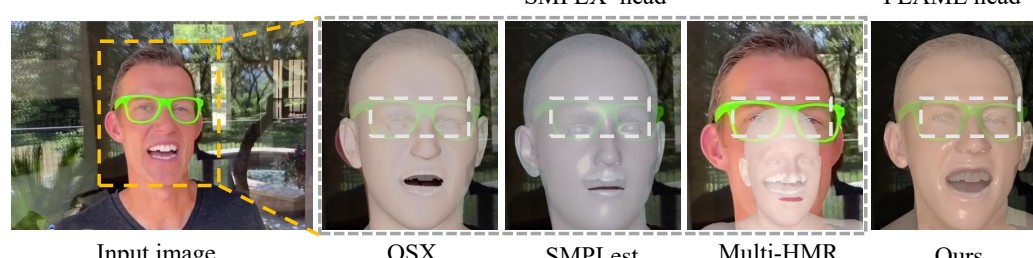

Figure 2: **SMPLX vs. FLAME:** In head modeling, SMPLX provides less expressive facial expressions than FLAME, while existing SMPLX-based methods are further limited by noticeable mesh–image misalignments.

low-dimensional controllable parameters, thus enabling direct regression of body parameters from single or multiple images and significantly advancing the development of 3D human pose estimation.

However, although these human models excel at capturing global body structure and large-scale poses, they are limited in modeling fine-grained details. In particular, they lack sufficient degrees of freedom to accurately represent subtle facial expressions, as shown in Fig. 2, making it difficult to convey rich emotional and interactive cues and thereby constraining practical applications. To address this limitation, GUAVA Zhang et al. (2025) proposes integrating the more expressive FLAME face model Li et al. (2017) with SMPLX Pavlakos et al. (2019a) by replacing the corresponding head vertices, which preserves the ability to model the full body while substantially enhancing facial expressiveness. GUAVA refers to this extended formulation as the expressive human model (EHM), which enables more natural capture of facial dynamics in full-body mesh reconstruction and provides stronger expressiveness and consistency for downstream tasks.

Moreover, we observe that current Human Mesh Recovery (HMR) methods primarily emphasize the alignment of the overall body structure and joint positions during pose estimation, while often neglecting alignment in fine-grained regions. This limitation becomes particularly evident in the mapping between 2D images and reconstructed meshes (see Fig. 2), where local misalignments frequently occur. For instance, in detail-rich regions such as the hands and face, the predicted meshes often deviate noticeably from the input images. Such discrepancies not only undermine the realism and naturalness of the reconstructions but can also be amplified in downstream applications that rely heavily on fine-grained fidelity. The root cause is that existing methods mainly rely on body parameter regression and keypoint supervision. Such supervision provides insufficient pixel-level constraints, making it difficult to achieve fine-grained alignment between the image and the human mesh. To mitigate this problem, prior works Moon et al. (2024a); Zhang et al. (2025) attempt to align the human mesh with the corresponding image through optimization-based strategies. However, these optimization-heavy approaches are computationally expensive, severely limiting their applicability in large-scale downstream tasks.

To address these issues, we propose Pixel-aligned Expressive Human Mesh Recovery (termed as **PEAR**), which builds on the recently introduced Expressive Human Model(EHM) and employs a transformer-based network to directly regress EHM parameters from a single image, enabling richer facial expression modeling. To further alleviate misalignments in fine-grained regions—often arising from supervision limited to joints and pose parameters—we integrate a neural renderer that introduces photometric loss for improved detail alignment. By jointly training both modules, our approach achieves more accurate human parameter estimation and higher-fidelity rendering.

Another key challenge of this work lies in the lack of image datasets annotated with EHM parameters. To address this, we decompose the EHM model into three components: body, face, and hand parameters. Specifically, we obtain pseudo ground-truth body supervision from the current datasets annotated with SMPL parameters, while pseudo ground-truth for the face and hands is derived through fitting algorithms Zhang et al. (2025). In summary, our contributions are as follows:

**(1)** Our model predicts human meshes that are more pixel-aligned with the input images, avoiding the severe misalignments commonly observed in prior methods.

**(2)** We propose the PEAR framework, which jointly estimates SMPLX and FLAME parameters to recover a more expressive human model.

**(3)** Our method requires no additional tracking or cropping operations and, for the first time, achieves real-time 3D human reconstruction and animation, greatly facilitating downstream tasks.

**(4)** We construct a large-scale dataset of facial and hand parameters, suitable for training both SMPLX and FLAME models, which will be released to the community.

## 2  RELATED WORK

**Human Pose and Shape Estimation.**  Human pose estimation from images is a well-studied problem with numerous applications Tome et al. (2019); Zhu et al. (2024); Lin et al. (2023); Xiang et al. (2019); Mehta et al. (2017a); Pavlakos et al. (2019b). Early optimization-based approaches (e.g., SMPLify-XPavlakos et al. (2019b)) estimate body parameters from a single image through iterative fitting, but these methods are often time-consuming. Human Mesh Recovery (HMR) Kanazawa et al. (2018) alleviates this limitation by directly regressing SMPL Loper et al. (2015) parameters with a CNN, significantly reducing inference time. This idea has inspired a series of follow-up methods such as SPIN Kolotouros et al. (2019) and PARE Kocabas et al. (2021), which further improved the accuracy of human pose estimation and extended parameter regression to models like SMPLX Pavlakos et al. (2019a) and MANO Romero et al. (2017). More recently, the emergence of ViT-based methods has led to notable gains in estimation accuracy. Following previous advances Lin et al. (2023); Baradel et al. (2024); Goel et al. (2023); Xia et al. (2025), we also adopt a transformer-based neural network for EHM regression.

**Human Appearance Reconstruction.**  Traditional human reconstruction methods Daněček et al. (2022); Chen et al. (2022); Yuan et al. (2022); Saito et al. (2019) have primarily focused on mesh reconstruction, covering various parts such as the body, face, and hands. CAR Liao et al. (2023), SITH Ho et al. (2024), and CanonicalFusion Shin et al. (2025) respectively reconstruct animatable human bodies from single-view and multi-view images. ICON Xiu et al. (2022) and ECON Xiu et al. (2023) reconstruct clothed humans through implicit and explicit normal-fusion approaches. In recent years, the emergence of neural radiance fields (NeRF) and 3D Gaussian splatting (3DGS) Kerbl et al. (2023) has inspired numerous efforts Weng et al. (2022); Zhao et al. (2023); Yuan et al. (2024); Hu et al. (2024b); Lei et al. (2024); Liu et al. (2024) to combine human appearance with template models to achieve more realistic 3D reconstructions. Methods such as GART Lei et al. (2024), GaussianAvatar Hu et al. (2024a) and ExAvatar Moon et al. (2024b) are typically trained per individual ID, and thus lack generalization ability. More recently, approaches including Human-LRM Weng et al. (2024), Human-Splat Pan et al. (2024), LHM Qiu et al. (2025), and GUAVA Zhang et al. (2025) have focused on human appearance modeling with generalization capability. We refer to this class of methods as Neural Renderers, which can provide pixel-level supervisory signals. Recently, this field remains highly active, with numerous outstanding works Sun et al. (2024); Patel & Black (2025); Stathopoulos et al. (2024); Wang et al. (2025); Shen et al. (2025); Shin et al. (2025), focusing on expressive human pose estimation and 3D human reconstruction.

## 3  METHOD

We propose PEAR to address two major limitations of existing approaches: the inaccurate localization of fine-grained human pose details and the lack of sufficient photometric supervision for self-reconstruction. Prior SMPLX–based methods primarily emphasize body pose accuracy, often overlooking regions with richer details such as the face and hands. Moreover, relying solely on keypoint and parameter losses leads to misalignment in fine-grained localization. To overcome these issues, we leverage input data to introduce photometric supervision, and separately estimate FLAME parameters for more expressive facial modeling, thereby enabling more accurate and expressive human parameter estimation.

### 3.1  PRELIMINARY: EXPRESSIVE HUMAN MODEL

EHM was first introduced by GUAVA Zhang et al. (2025) to address SMPLX's difficulty in capturing fine-grained facial expressions. Leveraging FLAME's strong performance in this regard, EHM (Expressive Human Model) replaces the SMPLX head with FLAME, enabling more accurate facial expression representation. In our work, we adopt EHM as the target human model for parameter prediction, allowing the mesh to faithfully capture facial expressions without the stiffness observed in SMPLX.

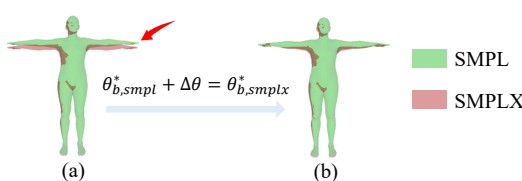

Figure 4: **Pipeline of PEAR.** PEAR is trained in two stages. In the first stage, we use single-frame images (employing only the target flow) to train a ViT-based architecture for estimating EHM parameters, including SMPLX body and FLAME head parameters. In the second stage, we use image pairs (from video datasets) as training units and incorporate a neural renderer to provide photometric supervision, thereby enabling joint training of the two modules and enhancing human detail reconstruction and rendering quality.

## 3.2 EXPRESSIVE HUMAN MESH ESTIMATION

**Architecture.** Our model architecture follows the design used in HMR2 Goel et al. (2023), employing a ViT backbone that takes a single RGB image as input. At the network output, multiple transformer heads are attached to regress SMPLX, FLAME, and camera parameters. While this architecture has previously been validated primarily for accurate body pose estimation Goel et al. (2023), our experiments demonstrate that **it is also capable of reconstructing fine-grained facial and hand details from low-resolution images** without requiring body-part cropping, thereby eliminating time-consuming preprocessing.

Figure 3: Differences between the SMPLX and SMPL human models.

For the model outputs, we primarily regress two sets of parameters of EHM. **SMPLX**: pose parameters $\theta_b$ and shape parameters $\beta_b$. **FLAME**: pose parameters $\theta_h$, shape parameters $\beta_h$, and expression parameters $\phi_h$. The above process can be formulated as follows: given an input image $I$, the human pose model $F_{vit}$ predicts $[\theta_b, \beta_b, \theta_h, \beta_h, \phi_h, \pi] = F_{vit}(I)$. Finally, the mean squared error (MSE) losses are applied to the SMPLX parameters:

$$\mathcal{L}_{body} = ||\theta_b - \theta_b^*||_2^2 + ||\beta_b - \beta_b^*||_2^2, \tag{1}$$

where $\theta_b^*$ and $\beta_b^*$ denote the ground-truth SMPLX pose parameters and shape parameters, respectively. For supervision of the FLAME parameters, we employ an $L_1$ loss:

$$\mathcal{L}_{head} = ||\theta_h - \theta_h^*||_1 + ||\beta_h - \beta_h^*||_1 + ||\phi_h - \phi_h^*||_1. \tag{2}$$

Here, $\theta_h^*$, $\beta_h^*$, and $\phi_h^*$ represent the ground-truth FLAME pose, shape, and expression parameters, respectively. We also incorporate 3D and 2D keypoints as supervision, using the L1 loss:

$$\mathcal{L}_{kp} = ||X - X^*||_1 + ||\pi(X) - x^*||_1, \tag{3}$$

where $X^*$ and $x^*$ denote the ground-truth 3D and 2D keypoints, respectively, and $X$ represents the predicted 3D keypoints. $\pi(X)$ denotes the projection of the 3D points onto the image plane using camera parameters $\pi$. The 2D keypoints $x^*$ include both body and facial landmarks. The 3D keypoint loss is computed only for samples with available 3D ground truth.

**EHM Training Data Generation.** One of the key challenges in training the expressive human pose model $F_{Vit}$ is the absence of image datasets that provide both SMPLX and FLAME annotations. To ensure accurate pose estimation for different body parts, we divide the human body into three components: body, hands, and face. For the body part, we leverage the SMPL annotations from existing datasets as pseudo ground-truth and adapt them into SMPLX body parameters with slight

adjustments. As shown in Fig. 3, directly replacing SMPL body poses parameters with SMPLX body poses results in inconsistencies due to an inherent offset between the two human models. To address this issue, we compute an offset $\Delta\theta$ through optimization to align their T-poses, thereby enabling the SMPLX body pose parameters to be directly derived from datasets annotated with SMPL. This can be formulated as

$$\theta_{b,smplx}^* = \theta_{b,smpl}^* + \Delta\theta, \tag{4}$$

where $\theta_{b,smplx}^*$ and $\theta_{b,smpl}^*$ represent the body pose parameters of SMPLX and SMPL, respectively. For the hands and face, we adopt the HAMER Pavlakos et al. (2024) and TEASER Liu et al. (2025) models to coarsely estimate the SMPLX hand pose parameters, as well as the FLAME facial shape $\beta_h^*$, expression $\phi_h^*$, and pose $\theta_h^*$. Subsequently, we extract body keypoints $x_b^*$ and facial keypoints $x_h^*$ using keypoint detection models, which are further refined following the strategy introduced in GUAVA Zhang et al. (2025).

### 3.3 Pixel-level enhancement

We observe that supervision solely based on body parameters and keypoints is insufficient for optimizing fine-grained regions of the human body. To address this, we further introduce a photometric loss to provide pixel-level supervision, ensuring that the predicted human model is more accurate in detailed regions and better aligned with the image pixels. Specifically, for video data, we randomly select two frames as the source and target, and feed them into the human pose estimation model $F_{vit}$ to extract the human body model parameters as follows:

$$[\theta_b^s, \beta_b^s, \theta_h^s, \beta_h^s, \phi_h^s, \pi^s], [\theta_b^t, \beta_b^t, \theta_h^t, \beta_h^t, \phi_h^t, \pi^t] = F_{vit}(I_s), F_{vit}(I_t). \tag{5}$$

Subsequently, we combine the two sets of parameters into $\Phi = [\theta_b^t, \beta_b^s, \theta_h^t, \beta_h^s, \phi_h^t, \pi^t]$ based on which the final reconstructed image can be expressed as:

$$\hat{I}_t = F_{ren}(F_{ehm}(\Phi), I_s, \pi^t). \tag{6}$$

Here, $F_{ehm}$ and $F_{ren}$ denote the expressive human model and the neural renderer, respectively, where we adopt GUAVA Zhang et al. (2025) as the neural renderer. With this formulation, we can seamlessly introduce the photometric loss to provide pixel-level supervision:

$$\mathcal{L}_{photo} = \mathcal{L}_1(I_t, \hat{I}_t) + \mathcal{L}_{lpips}(I_t, \hat{I}_t). \tag{7}$$

By jointly training the two components, we achieve mutually reinforcing improvements: the human pose estimation model $F_{vit}$ attains higher accuracy in human parameter estimation, while the neural renderer benefits from enhanced rendering performance, as evidenced by the results in Tab. 4.

### 3.4 Implementation Details

In summary, our training pipeline consists of two stages. **In the first stage,** we train the ViT-based model $F_{vit}$ on large-scale *image datasets* to estimate the parameters of the EHM model. This stage is conducted for approximately 200k iterations with a batch size of 320 on 8 NVIDIA A6000 GPUs, taking about 7 days. **In the second stage**, we incorporate a neural renderer to introduce the photometric loss $\mathcal{L}_{photo}$, which further refines fine-grained human body details and enforces pixel-level alignment of the predicted EHM model. This stage is trained on *video datasets* for roughly 20k iterations with a batch size of 16 on 8 A6000 GPUs, requiring about 1 day.

## 4 Experiment

In this section, we conduct both qualitative and quantitative evaluations of our human reconstruction and rendering framework. First, compared with previous SMPLX-based approaches Zhang et al. (2023); Lin et al. (2023); Yin et al. (2025); Baradel et al. (2024), we demonstrate that PEAR not only exhibits strong generalization ability, enabling accurate human model estimation under diverse and complex environments, but also achieves more precise pose estimation in fine-grained body regions with improved pixel-level alignment (Sec. 4.2). Second, we further show the superior performance of our method in rendering-driven downstream applications (Sec. 4.3), highlighting its robustness and accuracy in human pose recovery. Finally, we present ablation studies to validate the effectiveness of our approach (Sec. 4.4).

Table 1: Quantitative comparison of human heads.

| Methods | UBody (GUAVA) | | 3DPW | |
|---|---|---|---|---|
| | MLE↓ | LVE↓ | MLE↓ | LVE↓ |
| | ($\times 10^{-3}$ m) | ($\times 10^{-5}$ m) | ($\times 10^{-3}$ m) | ($\times 10^{-4}$ m) |
| SMIRK | 2.81 | 8.02 | 4.25 | 2.77 |
| TEASER | 1.92 | 4.23 | 3.95 | 5.60 |
| Ours | **0.99** | **2.55** | **1.93** | **0.74** |

Table 2: Quantitative comparison of human hands (PA-PVE) on the UBody (OSX) and EHF test dataset.

| Method | Backbone | Reso. | EHF↓ | UBody↓ |
|---|---|---|---|---|
| PIXIE Feng et al. (2021) | RN50 | crop | 11.1 | 12.2 |
| Hand4Whole Moon et al. (2022) | RN50 | crop | 10.8 | 8.9 |
| OSX Lin et al. (2023) | ViT-L/16 | $256 \times 192$ | 15.9 | 10.8 |
| SMPLer-X Cai et al. (2023) | ViT-L/16 | $256 \times 192$ | 15.0 | 10.3 |
| Multi-HMR Baradel et al. (2024) | ViT-B/14 | $896 \times 896$ | **12.2** | **8.8** |
| Ours | ViT-B/16 | $256 \times 192$ | 12.8 | **8.8** |

Table 3: Quantitative results on PCK. Our method achieves more accurate human pose estimation compared to SMPLX-based approaches. *PyMAF-X trained on LSP dataset.

| Methods | Model | COCO | | LSP | | PoseTrack | |
|---|---|---|---|---|---|---|---|
| | | @0.05↑ | @0.1↑ | @0.05↑ | @0.1↑ | @0.05↑ | @0.1↑ |
| CLIFF Li et al. (2022) | SMPL | 0.64 | 0.88 | 0.32 | 0.66 | 0.75 | 0.92 |
| PARE Kocabas et al. (2021) | SMPL | 0.72 | 0.91 | 0.27 | 0.66 | 0.79 | 0.93 |
| HMR2 Goel et al. (2023) | SMPL | 0.87 | 0.97 | 0.53 | 0.82 | 0.90 | 0.98 |
| HSMR Xia et al. (2025) | SKEL | 0.87 | 0.96 | 0.51 | 0.81 | 0.90 | 0.98 |
| PyMAF-X* Zhang et al. (2023) | SMPLX | **0.79** | 0.93 | - | - | 0.85 | 0.95 |
| OSX Lin et al. (2023) | SMPLX | 0.70 | 0.87 | 0.42 | 0.73 | 0.82 | 0.90 |
| SMPLest-X Yin et al. (2025) | SMPLX | 0.71 | 0.91 | 0.40 | 0.74 | 0.82 | 0.91 |
| Ours | EHM | **0.79** | **0.94** | **0.52** | **0.80** | **0.87** | **0.97** |

## 4.1 SETUP

**Dataset.** In the first stage, following prior work, we train on our processed large-scale image datasets: Human3.6M Ionescu et al. (2014), MPI-INF-3DHP Mehta et al. (2017b), COCO Lin et al. (2014), MPII Andriluka et al. (2014), InstaVariety Kanazawa et al. (2019), AVA Gu et al. (2018), and AI Challenger Wu et al. (2017). In the second stage, we train on video datasets, including Ubody (GUAVA) Zhang et al. (2025) and Seamless Interaction Agrawal et al. (2025).

**Baseline.** We report results on benchmarks commonly used for comparison with a wide range of prior methods. Since our approach integrates both SMPLX and FLAME models, we evaluate against SMPL- and SMPLX-based methods for body and hand reconstruction, and against dedicated FLAME-based methods for face reconstruction.

## 4.2 POSE ACCURACY

**Head evaluation metrics.** For 3D head mesh modeling, we assess reconstruction accuracy using key metrics such as lip vertex error (LVE) Richard et al. (2021) and mean vertex error (MVE), which measure the deviation of mouth and overall facial vertices from the tracking results. Since no dedicated benchmark exists, we follow the fitting strategy proposed in GUAVA to generate test samples from the 3DPW von Marcard et al. (2018) and UBody Zhang et al. (2025) test splits. Quantitative comparisons with SOTA learning-based methods TEASER and SMIRK (Tab. 1) shows that our method captures high-quality facial expressions without face cropping, comparable to theirs.

**Hand evaluation metrics.** Following prior works, we evaluate the Procrustes Alignment per-vertex error (PA-PVE) metric for hands on the EHF and UBody-intra Lin et al. (2023) datasets, as shown in Tab. 2. Among the baselines, PIXIE Feng et al. (2021) and Hand4Whole Moon et al. (2022) require additional cropping of the hand region, while Multi-HMR Baradel et al. (2024) uses an input resolution of $896 \times 896$, which results in lower efficiency. In contrast, our method only requires an input resolution of $256 \times 256$ and achieves comparable performance without any extra processing.

**Body keypoint metrics.** We evaluate 2D image alignment of the generated human poses by reporting Percentage of Correct Keypoints (PCK) of reprojected keypoints at different thresholds as shown in Tab. 3. Our method performs slightly worse than specialized body pose estimation approaches such as HSMR and HMR2, since they only focus on simple body pose without modeling complex facial and hand details. Nevertheless, our approach significantly outperforms other SMPLX-based methods. Fig. 7 shows PEAR under more complex poses.

**Human mesh visualization.** As shown in Fig. 5, our method captures richer facial details and achieves better pixel-level alignment on both UBody and WholeBody images, demonstrating strong

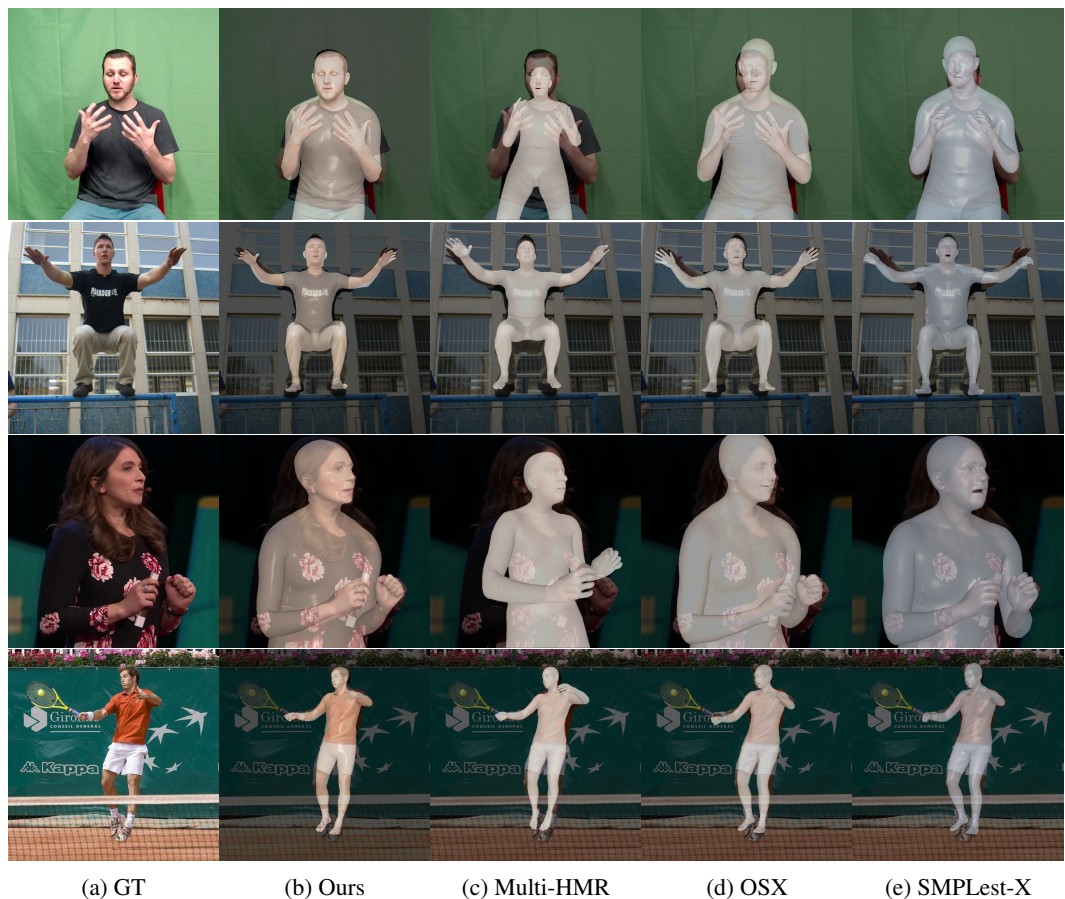

(a) GT      (b) Ours      (c) Multi-HMR      (d) OSX      (e) SMPLest-X

Figure 5: Qualitative results. We compare PEAR with several smplx-based sota approaches.

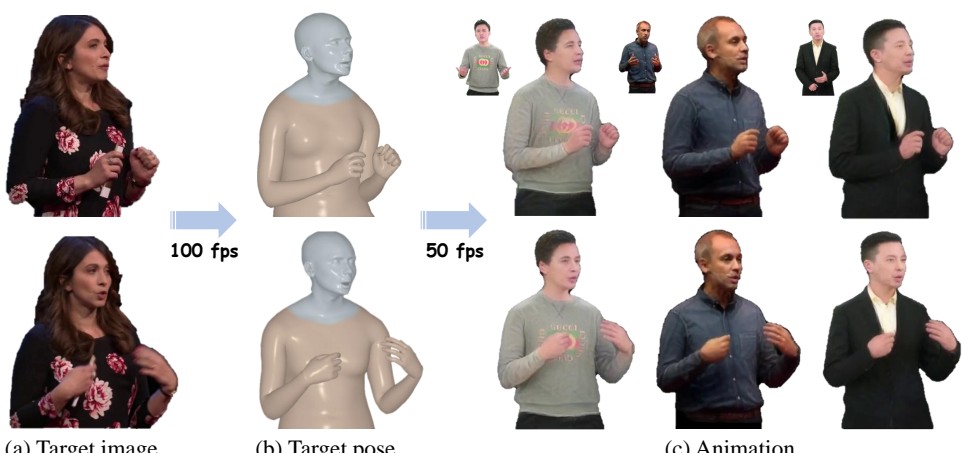

(a) Target image      (b) Target pose      (c) Animation

Figure 6: Downstream Applications. Unlike previous approaches, our method does not require explicit cropping of the face or hands, and can achieve high-quality human pose estimation from a single low-resolution image. This provides a faster interface for downstream applications.

generalization in human pose estimation. On the other hand, unlike OSX and SMPLest-X, our model does not require explicit cropping of the face and hands, nor does it need high-resolution input images like Multi-HMR. It can accurately infer parameters for all human body parts from a single low-resolution image of 256×192 in one feed-forward pass, saving more than $10\times$ the inference time, providing a faster interface for downstream applications.

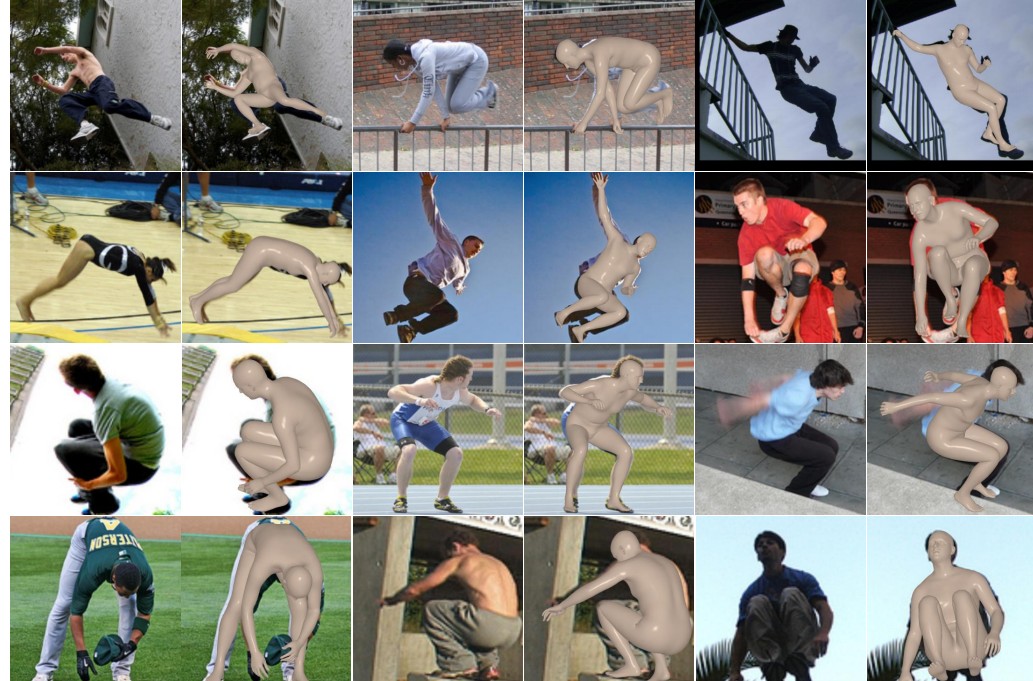

Figure 7: We further visualize the performance of our model on the LSP-Extend dataset, which poses significant challenges for human pose estimation methods.

Table 4: Quantitative results. *Tracking* denotes EHM parameter estimation via an optimization-based approach Zhang et al. (2025). *M1/R1* and *M2/R2* refer to the human pose model $F_{vit}$ and neural renderer $F_{ren}$ before and after stage 2 (pixel-level enhancement). *Time 1* indicates human pose estimation per image, and *Time 2* the performance driving time (excluding human reconstruction time 0.1s).

| Type | EHM Param. | Renderer | PSNR↑ | SSIM↑ | LPIPS↓ | Time 1↓ | Time 2↓ | Total Time↓ |
|------|-----------|----------|-------|-------|--------|---------|---------|-------------|
| A | Tracking | R1 | 24.68 | 0.892 | 0.0824 | 2 min | 0.01s | 2 min |
| B | M1 | R1 | 24.15 | 0.882 | 0.0883 | - | - | - |
| C | M2 | R1 | 25.36 | 0.898 | 0.0793 | - | - | - |
| D | M2 | R2 | **25.50** | **0.901** | **0.0784** | 0.01s | 0.01s | **0.02s** |

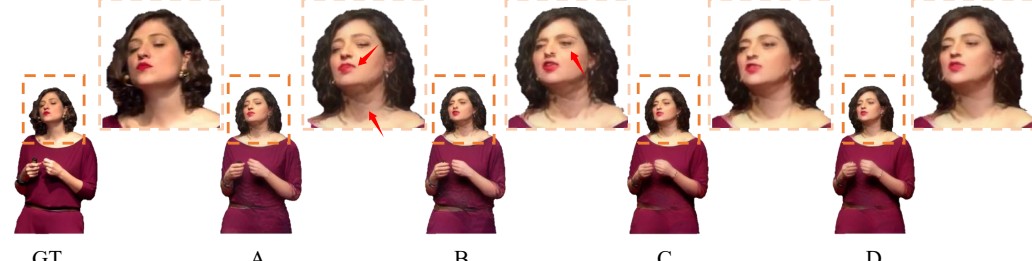

GT      A      B      C      D

Figure 8: Qualitative results: direct concatenation of the two models yields suboptimal performance (Type B), while joint training improves outcomes (Types C and D).

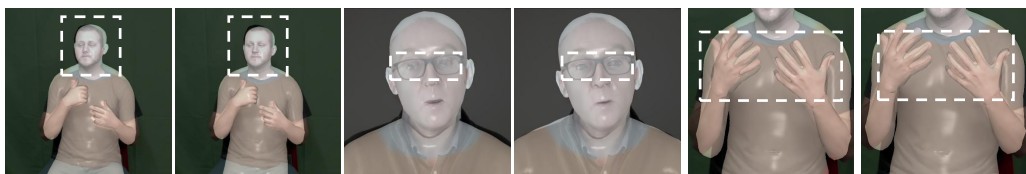

Figure 9: Ablation on two-stage training. Joint training improves fine-grained body part alignment (left: w/o joint training, right: w joint training).

Table 5: Ablation study results (in meters) on 4 different datasets. MLE $(\times 10^{-4})$ and LVE $(\times 10^{-5})$ measure facial expression accuracy; PA-PVE measures hand vertex accuracy. Stage 1: $F_{vit}$ training; Stage 2: pixel-level enhancement.

| Methods | UBody | | 3DPW | | EHF | UBody-intra |
|---|---|---|---|---|---|---|
| | MLE↓ | LVE↓ | MLE↓ | LVE↓ | PA-PVE↓ | PA-PVE↓ |
| stage 1 | 11.0 | 2.72 | 6.21 | 1.10 | 13.3 | 9.5 |
| + stage 2 | **9.92** | **2.55** | **1.93** | **0.74** | **12.8** | **8.8** |

### 4.3 RENDERING QUALITY

We incorporate a neural renderer Zhang et al. (2025) into our framework, not only to provide pixel-level supervision but also to enable mutually reinforcing improvements through joint training of the two modules. To validate this, we conduct cross experiments using widely adopted metrics, including PSNR, SSIM, and LPIPS. As shown in Tab. 4 and Fig. 8, type B, simply concatenating the two modules (the human pose estimation model $F_{vit}$ and the neural renderer $F_{ren}$) sequentially yields suboptimal results, whereas type C and D demonstrate that joint training improves the rendering performance of both modules.

Moreover, the EHM parameters predicted by our model achieve higher rendering quality than those obtained through tracking Zhang et al. (2025). This demonstrates that our approach enables real-time, second-scale human performance driving, as shown in Fig. 6, without requiring parameter tracking as in methods such as LHM Qiu et al. (2025) and GUAVA Zhang et al. (2025).

### 4.4 ABLATION

We conduct comprehensive ablation studies on the proposed two-stage training scheme. First, we examine the impact on rendering quality, as reported in Tab. 4 and Fig. 8. The results show that directly concatenating the two modules yields only limited performance, whereas joint training enables the modules to complement each other, leading to noticeable improvements in rendering fidelity. Beyond rendering, we further evaluate the accuracy of human body reconstruction. As shown in Tab. 5 and Fig. 9, the joint training strategy significantly enhances the recovery of fine-grained details, particularly in regions such as the hands and face. These improvements confirm our hypothesis that stage-2 pixel-level supervision effectively enforces a tighter alignment between the input images and the reconstructed meshes, thereby producing more realistic results.

## 5 DISCUSSION

One of our primary goals is to address the pixel-level misalignment issue that is prevalent in existing HMR approaches. This problem not only undermines the realism and naturalness of the reconstructions but also becomes particularly detrimental in downstream applications that demand high precision. To alleviate this, we incorporate a large amount of training data with only 2D annotations. However, such supervision often leads to degraded 3D evaluation metrics (e.g., MPJPE). We attribute this to the inherent ambiguity of recovering 3D pose from 2D images: a single human image may correspond to multiple plausible 3D poses. Introducing additional 2D annotations as supervision biases the predicted human poses toward fitting the 2D labels, leading to a deviation from the 3D annotations. Nevertheless, the 3D poses predicted by our model remain plausible. This phenomenon has also been validated in HMR2 Goel et al. (2023).

## 6 CONCLUSION

In this paper, we present PEAR, the first human mesh recovery framework that simultaneously regresses both SMPLX and FLAME parameters, addressing two key limitations of prior approaches. First, previous HMR methods often fail to achieve pixel-level alignment in the image plane, leading to misaligned human meshes and constraining their applicability in high-precision downstream tasks. Second, SMPLX exhibits limited capacity in modeling facial details, making it inadequate for representing diverse expressions. To overcome these challenges, PEAR takes a single image as input and jointly estimates SMPLX and FLAME parameters, thereby producing more expressive human meshes. Overall, our method elevates the accuracy of HMR to the pixel level and significantly enhances the fidelity and applicability of human mesh representations in downstream applications.

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

# A APPENDIX

## A.1 USE OF LLMS

We only used LLMs as a language polishing tool, without involving them in method design, experimental design, or any other aspects.

## A.2 ETHICS STATEMENT

This research project has been approved by the relevant ethics committee or institution and has been conducted in strict accordance with ethical guidelines. The rights and privacy of participants were respected and protected, and personal information was kept confidential.

1. Informed Consent: All participants were informed of the purpose, procedures, risks, and benefits of the study, either verbally or in writing, and their informed consent was obtained.

2. Data Confidentiality and Privacy Protection: Appropriate measures were taken to safeguard participants' personal information and privacy.

3. Use of Research Data: The use of research data adhered strictly to principles of legality and transparency, ensuring proper use and interpretation of the data.

## A.3 REPRODUCIBILITY STATEMENT

We ensure that our method is fully reproducible, and we will publicly release the training data, code, and model weights upon paper acceptance.

**For more visual results, please refer to the supplementary material we provide.**

## A.4 MORE EXPERIMENTS

Following Reviewer fZ77 and Reviewer 7qEi, we have added more examples of extreme cases, such as extreme expressions, lighting variations, motion blur, and occlusions, to give readers a better understanding of the limitations of our method.

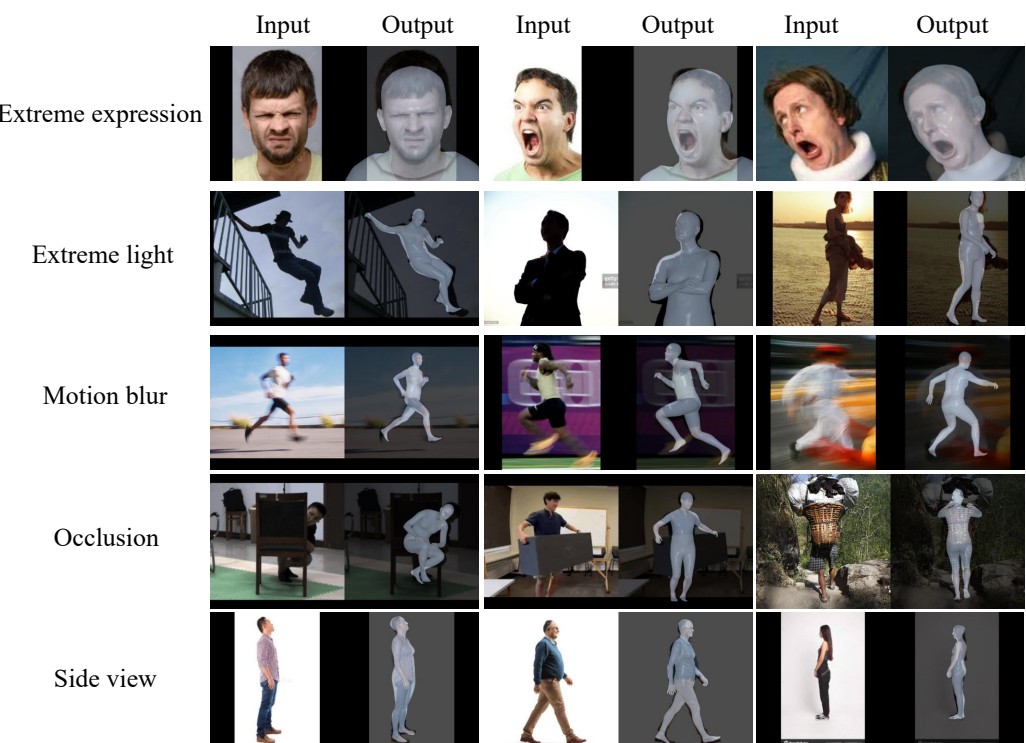

Figure 10: **Extreme cases.**

| Method | Backbone | Image Processed | Body ↓ | Hand ↓ | Infer Time ↑ |
|--------|----------|-----------------|--------|--------|--------------|
| Hand4Whole | RN50 | Crops | 90.2 | 47.2 | - |
| PyMAF-X | HRNet48 | Crops | 84.0 | 45.1 | - |
| Multi-HMR | ViT-L | 896×896 | - | 40.7 | 5 FPS |
| Ours | ViT-B | 256×192 | 81.9 | 41.2 | 100 FPS |

Table 6: MVE evaluation on the AGORA dataset

