# OpenReview forum: "PEAR: Pixel-aligned Expressive humAn mesh Recovery"
_ICLR.cc/2026/Conference — ICLR 2026 Conference Withdrawn Submission_

### Official Review · Reviewer_Togf · 2025-10-15

**Soundness:** 3
**Presentation:** 3
**Contribution:** 3
**Rating:** 6
**Confidence:** 5

**Summary:**

This paper proposes PEAR (Pixel-Aligned Expressive Human Mesh Recovery), a unified framework that recovers expressive 3D human meshes from a single image. Unlike prior SMPL-X based methods, PEAR integrates the FLAME head into SMPL-X (forming the Expressive Human Model, EHM), improving facial expressiveness while preserving full-body modeling. The method further introduces a two-stage training pipeline:

* A ViT-based regressor predicts EHM parameters (body, face, hands, camera).

* A neural renderer is used with photometric loss for pixel-level enhancement.

Experiments on datasets such as UBody, 3DPW, EHF, and LSP-Extended show improvements in facial detail, hand accuracy, and overall pixel alignment. The method also demonstrates real-time avatar reconstruction (0.05s inference) and generalizes to downstream animation tasks. Ablation studies validate the benefit of two-stage training.

**Strengths:**

1. Pixel-level photometric supervision: The second-stage neural rendering significantly improves fine-grained alignment beyond joint/parameter losses.

**Weaknesses:**

1. FLAME head pose integration:
The paper states the proposed system estimates both SMPL-X and FLAME parameters, but it is unclear how global head orientation is consistently maintained. For example, when the body is rotated or facing away, naïvely replacing the head could cause inconsistencies between the body and face orientation. Clarification is needed on how alignment between body root pose and FLAME global pose is enforced.

2. Stage-2 reliance on upper-body datasets:
The second-stage training leverages upper-body 3DGS datasets (e.g., UBody) for pixel-level enhancement. This raises two concerns:

* Lower-body accuracy may degrade since pixel-level supervision is not applied consistently to the legs/feet.

* Upper-body datasets typically lack diverse whole-body poses, potentially limiting generalization.
Related to this, Table 5 shows ablation across UBody, 3DPW, EHF, and UBody-intra, but it is not clearly explained how the proposed method addressed the above two concerns.

3. Limited full-body evaluations:
While the paper reports on 3DPW, EHF, and UBody, it lacks broader validation on challenging full-body datasets like AGORA or BEDLAM, which test both pixel alignment and kinematic reconstruction under more diverse settings. Without these, the claim of strong generalization to complex whole-body scenarios is not fully convincing.

**Questions:**

Please refer to Weaknesses

---

> ### Author Response · Authors · 2025-11-15
>
> Thank you for your recognition of our work. We will address your concerns carefully.
>
> ### **Q1. FLAME Head Pose Integration: It remains unclear how global head orientation consistency is maintained across the two models.**
>
> During the forward pass, we estimate only the global pose of SMPL-X, leaving the global pose of FLAME unestimated. For fusion, we directly replace the SMPL-X head mesh with the FLAME head mesh, addressing the orientation consistency problem. To ensure seamless integration, we also estimate a scale factor to minimize size discrepancies between the SMPL-X body and the FLAME head.
>
> ### **Q2. The dependence on upper-body datasets in the second stage may lead to decreased accuracy and generalization performance for the lower body.**
>
> In the second-stage training, we use two datasets—the full-body Seamless Interaction [1] and the upper-body Ubody [2]—and adopt a slow–fast training strategy that leaves lower-body performance largely unaffected while substantially enhancing the modeling of facial and hand details. Moreover, **this stage consists of only a 20k-step fine-tuning process**, ensuring that the model’s overall generalization ability remains largely preserved. The full evaluation is presented below:
>
>
> |   |   COCO (0.05/0.1)$\uparrow$ | PTrack (0.05/0.1)$\uparrow$ | UBody (MLE/LVE)$\downarrow$  | 3DPW (MLE/LVE)$\downarrow$  | EHF (PAPVE)$\downarrow$  | UBody(PAPVE)$\downarrow$  |
> |:--------:|:-----------------:|:---------------------:|:---------------:|:--------------:|:-----------:|:-----------------:|
> | stage1   | **0.79 / 0.94**       | **0.87 / 0.97**           | 11.0 / 2.72     | 6.21 / 1.10    | 13.3        | 9.5               |
> | +stage2  | 0.78 / **0.94**       | 0.86 / 0.96           | **9.92 / 2.55**     | **1.93 / 0.74**    | **12.8**        | **8.8**               |
>
> The first two metrics evaluate body joint accuracy, the middle two correspond to facial performance, and the last two measure hand accuracy. As shown, adding the second training stage does not significantly affect the model’s generalization ability, while providing a stronger improvement for the latter two regions. **This reflects an inherent trade-off: slightly reducing the model’s sensitivity to body limbs allows it to better capture the richer information present in the face and hands.** To verify this, please refer to our updated supplementary materials, where we provide real-time results of our model running in real-world scenarios.
>
> ### **Q3: Limited full-body evaluation: Although experiments were conducted on 3DPW, EHF, and UBody, validation on more challenging full-body datasets (e.g., AGORA or BEDLAM) is lacking.**
>
> Regarding strong generalization in complex full-body scenarios, following HMR2 and HSMR, we evaluated the accuracy of body keypoints on the full-body datasets COCO, LSP, and PoseTrack (Tab.3 in the main paper), which provide highly challenging and practical real-world validation. For BEDLAM, many methods use it as a training dataset rather than for testing, so we did not find a unified and fair evaluation protocol to compare the performance of different approaches. As for AGORA, the MVE evaluation is as follows:
>
> | Method       | Backbone | Image Processed | Body $\downarrow$  | Hand $\downarrow$|  Infer time $\uparrow$ |
> |:------------:|:--------:|:---------------:|:------:|:--------:|:--------:|
> | Hand4Whole   | RN50     | Crops       | 90.2   | 47.2     |  - |
> | PyMAF-X      | HRNet48  | Crops       | 84.0   | 45.1     |  - |
> | Multi-HMR    | Vit-L    | 896 × 896       | -      | 40.7     |  5 FPS |
> | Ours         | Vit-B    | 256 × 192       | 81.9   | 41.8     | 100 FPS|
>
> PEAR may fall slightly behind Multi-HMR on full-body pose estimation due to the difference in input resolution. However, it exhibits much stronger generalization ability and can robustly **handle cases where only the head or upper body is visible**, as shown in our supplementary materials. Moreover, **PEAR achieves an inference speed of 100 FPS**, providing a fast and stable interface for a wide range of downstream applications.  We invite you to review our updated supplementary materials, which include the **performance of our model in real-world scenarios**—estimating human pose from real-time video streams and driving and rendering avatars.
>
>
> ### **Thank you for your recognition of our work. Following your suggestion, we have revised the paper. Due to page limitations, we placed the experiments in the supplementary materials and will reorganize the layout in a later revision. If anything remains unclear, we would be happy to discuss it further.**
>
>
>
> ### **Reference**
>
> [1] Seamless interaction: Dyadic audiovisual motion modeling and large-scale dataset.
>
> [2] GUAVA: Generalizable upper body 3d gaussian avatar.

---

> > ### Comment · Reviewer_Togf · 2025-11-18
> >
> > Thanks for the response.
> > However, the authors’ replies still do not fully address my concerns.
> >
> > 1.
> > The explanation regarding FLAME is still unclear. If the global pose of FLAME is not estimated, how exactly is the FLAME mesh aligned when replacing the face region of the SMPL-X mesh? Moreover, what happens when the face is not visible? In such cases, the FLAME output may have a global pose inconsistent with the full-body context, and it is unclear how the method resolves this.
> >
> > 2.
> > The authors report only 2D metrics for body evaluation. Without any 3D evaluation, it is difficult to judge whether the second stage degrades the 3D body accuracy or not.
> >
> > 3.
> > It is also unclear where the performance gains come from. As far as I understand, aside from the photometric loss, most architectural components, loss terms, and training datasets are similar to HMR2.0. The authors should clarify which part of the proposed method is responsible for the performance improvements.

---

> ### Author Response · Authors · 2025-11-19
> **Author Response to Reviewer Togf**
>
> We sincerely appreciate your thorough and insightful review. Your questions have helped us clarify important technical details and strengthen our paper. We are enthusiastic about addressing your concerns and demonstrating the robustness of our approach.
>
> > ### **Q1: If the global pose of FLAME is not estimated, how exactly is the FLAME mesh aligned when replacing the face region of the SMPL-X mesh?**
>
> The alignment process proceeds as follows:
> 1. We obtain the refined shape and expression-aware head mesh by passing shape and expression parameters through FLAME.
> 2. We obtain the SMPL-X template based on SMPL-X's shape and expression parameters.
> 3. In SMPL-X's canonical space, we perform registration/alignment based on joints, replace the corresponding head mesh, and obtain the SMPL-X template with the replaced head.
> 4. Finally, we update the template from the previous step through LBS to obtain the final mesh.
>
> The pseudo-code is as follows:
>
> ```pseudo
> # Step 1: Compute FLAME head mesh via LBS
> (head_vertices, head_joints) ← LBS(head_pose, template_head_vertices, ...) # global_pose removed
>
> # Step 2: Align FLAME head to SMPL-X body
> head_center ← Mean(head_joints[3 : 5])
> body_center ← Mean(body_joints[23 : 25])
> aligned_head ← head_vertices − head_center + body_center
>
> # Step 3: Replace SMPL-X head region with FLAME head
> template_body_vertices[smplx2flame_indices] ← aligned_head
>
> # Step 4: Final LBS to obtain fused full-body mesh
> (body_vertices, body_joints) ← LBS(body_pose, global_pose, template_body_vertices, ...)
> ```
>
> > ### **Q2:  What happens when the face is not visible?**
>
> When the face is not visible, the model tends to predict default values (no expression). Visualization results can be viewed on  [image](https://pear2025.github.io/PEAR-anonymous/static/images/extreme_case.png).
>
> > ### **Q3:  The FLAME output may have a global pose inconsistent with the full-body context, and it is unclear how the method resolves this**
>
> We replace the SMPL-X head with the FLAME head in the T-pose configuration, and then perform LBS and rotation transformations together. Therefore, there is no inconsistency issue.
>
> > ### **Q4:  Without any 3D evaluation, it is difficult to judge whether the second stage degrades the 3D body accuracy or not.**
>
> The concern you raised is very valuable. To verify whether the second stage affects 3D body accuracy, we conducted a rigorous comparison between Stage 1 and Stage 2 on the standard 3D benchmark (3DPW). The results are shown in the table below.
>
> | | 3DPW (MPJPE/PA-)$\downarrow$ |
> |:-:|:-:|
> | stage1 | 91.2 / 62.5  |
> | +stage2| 90.2 / 59.4 |
>
> As can be seen, Stage 2 brings only very minor numerical changes, indicating that the second-stage photometric finetuning does not compromise 3D structure or cause significant accuracy degradation.
>
> The reasons are as follows:
> 1. The training data used in Stage 2 has reliable 3D supervision, just like Stage 1, so the network's 3D prediction capability is not biased by the photometric loss.
> 2. The photometric loss is designed as a lightweight detail compensation, with its weight much lower than the 3D loss, so it does not dominate the optimization process.
> 3. We have verified in experiments that as long as reasonable **loss weights and finetuning epochs** are maintained, Stage 2's pixel-level alignment does not have a substantial impact on 3D error.
>
> In summary, the second stage brings significant geometric and pixel-level detail enhancement while maintaining 3D body accuracy essentially unchanged, which is exactly the purpose of our two-stage architecture design.
>
> > ### **Q5:  The authors should clarify which part of the proposed method is responsible for the performance improvements**
>
> The promising results of our method stem from the following key factors:
> 1. We introduce a novel learning paradigm for HMR methods by adopting an analysis-by-synthesis approach. Specifically, we incorporate a neural renderer that compares HMR outputs with rendered images, enabling the model to learn more accurate 3D human body representations.
> 2. From a data perspective, unlike previous SMPL-X methods, we redesign the data processing pipeline and separately annotate pseudo ground-truth for the body, face, and hands. This allows PEAR to achieve excellent initialization in the first stage of training.
> 3. In terms of architecture selection, we choose ViT-B as the backbone and a neural renderer that balances both speed and quality, enabling our method to achieve excellent performance in both aspects.
> 4. Regarding training, we introduce a denser photometric loss compared to previous HMR methods that rely on body keypoint supervision. This supervision provides more precise pixel-level guidance. Additionally, through our two-stage alternating training strategy, the models from both stages can mutually regularize each other, preventing overfitting and resulting in more accurate final results.
>
> Models, code, and data will be fully open-sourced. Thank you for your review!

---

### Official Review · Reviewer_7qEi · 2025-10-26

**Soundness:** 3
**Presentation:** 3
**Contribution:** 3
**Rating:** 4
**Confidence:** 4

**Summary:**

The paper introduces a Pixel-Aligned Expressive Human Mesh Recovery framework that aims to enhance fine-grained human mesh alignment and facial expressiveness from a single image. It jointly estimates SMPLX and FLAME parameters to form an Expressive Human Model (EHM), following GUAVA (ICCV 2025), addressing the common misalignment and expression limitations of existing human motion estimation works.

**Strengths:**

- Paper is well-written and easy to understand.
- By combining SMPLX and FLAME within the Expressive Human Model and using photometric supervision, the framework effectively captures subtle facial expressions and hand detail, outperforming recent works on human mesh recovery.

**Weaknesses:**

- The core design of this work that combines an EHM (SMPLX + FLAME) with a neural renderer for pixel-level photometric supervision,  closely follows the formulation of GUAVA (Zhang et al., ICCV 2025). While PEAR extends GUAVA’s upper-body focus to full-body reconstruction and adopts a two-stage training pipeline instead of optimization-based parameter tracking, the overall architecture and objective remain conceptually similar. The paper should clarify the key algorithmic differences or technical contributions beyond scaling the framework to full-body meshes.
- In Eq. (6), the reconstruction parameter set $\Phi = [\theta_t^b, \beta_s^b, \theta_t^h, \beta_s^h, \phi_t^h, \pi_t]$
 mixes body and facial parameters from different frames (source and target), specifically, shape parameters ($\beta$) from the source and other parameters from the target. No ablation or comparison is presented to validate that using source $\beta$ yields better alignment or rendering quality than other alternatives (e.g., all parameters from the target). Ablation study would be necessary to support this design.
- Although the “Human Appearance Reconstruction” part on Related work discusses neural renderers and Gaussian-based methods, it omits a large body of mesh-based literature that directly tackles high-fidelity human surface recovery from single or multiple images. Incorporating these works would better improve the quality of this paper:
    - [1] Saito et al., *“PIFu: Pixel-Aligned Implicit Function for High-Resolution Clothed Human Digitization,”* ICCV 2019.
    - [2] Shin et al., *“CanonicalFusion: Generating Drivable 3D Human Avatars from Multiple Images,”* ECCV 2024.
    - [3] Xiu et al., *“ICON: Implicit Clothed Humans Obtained from Normals,”* CVPR 2022.
    - [4] Xiu et al., *“ECON: Explicit Clothed Humans Optimized via Normal Integration,”* CVPR 2023.
    - [5] Liao et al., *“High-Fidelity Clothed Avatar Reconstruction from a Single Image,”* CVPR 2023.
    - [6] Ho et al., *“SITH: Single-view Textured Human Reconstruction with Image-Conditioned Diffusion,”* CVPR 2024.

I will reconsider score when all my concerns are handled well.

**Questions:**

How does the model perform under challenging conditions such as occlusions, extreme lighting, or side-view inputs? Including visual examples of such cases would help readers understand the framework’s limitations.

---

> ### Author Response · Authors · 2025-11-14
> **Author Response to Reviewer 7qEi (1/1)**
>
> We are very grateful for your detailed and constructive feedback. Your comments have helped us significantly strengthen the presentation of our technical contributions. The response of each point is as follows:
>
> > ### **Q1: The core design of this work ...  follows the formulation of GUAVA. The paper should clarify the key algorithmic differences or technical contributions beyond scaling the framework to full-body meshes.**
>
> These two methods are fundamentally different. Our method focuses on monocular HMR estimation, and by introducing an analysis-by-synthesis framework, **it can simultaneously and accurately predict face, body, and hand parameters at 100+ FPS.** The differences can be distinguished from the following aspects:
>
> 1. **Task difference**: Our method is a standard HMR task, aiming to estimate 3D human body parameters from a single image; whereas GUAVA is a human driving and reconstruction task, aiming to drive and generate new human images given a new 3D human model.
> 2. **Role of 3D human parameters**: The 3D human parameters that GUAVA relies on are obtained through offline processing, depending on complex and time-consuming fitting. Our method can estimate 3D human parameters from a single image at 100+ fps, making it applicable to other downstream tasks.
> 3. **Model architecture difference**: Our method designs a two-stage training framework with a ViT-based backbone, while GUAVA's architecture primarily fuses reference images with driven 3D human models and models 3DGS attributes to drive and generate human images.
> 4. **Dependency**: We use GUAVA as a neural renderer because it balances both speed and rendering quality. Architecturally, our method does not strictly depend on it, and other models capable of neural rendering can serve as alternatives.
>
> Furthermore, through PEAR's training, we can improve the rendering quality of the neural renderer (GUAVA), making the reconstruction results more realistic. Please refer to Fig. 8 and Tab. 4 for specific details.
>
> Additionally, we summarize our new perspectives to the HMR field as follows:
>
> 1. **Model architecture.** Prior SMPLX–based methods rely on high-resolution inputs or multi-branch designs for hands and face, resulting in slow inference. We are the first to show that a single lightweight ViT-B can jointly predict body, hand, and facial pose at over 100 FPS, enabling practical real-time deployment.
>
> 2. **Dataset perspective.** We redesign the annotation pipeline and build a large-scale, well-aligned dataset covering full-body, facial expressions, and hand poses across diverse viewpoints, lighting, occlusion, and motion. This dataset is essential for our method and provides a cleaner data-processing path for future research.
>
> 3. **Training strategy.** We propose a two-stage training framework: large-scale pretraining followed by pixel-level refinement using a differentiable neural renderer. This preserves strong generalization while substantially improving fine-grained fidelity.
>
> > ### **Q2:  PEAR extends GUAVA's upper-body focus to full-body reconstruction and adopts a two-stage training pipeline.**
>
> We need to clarify that PEAR is not an extension of GUAVA to full-body. GUAVA is a monocular Gaussian human reconstruction method, while PEAR is a fast full-body 3D parameter estimation method. Please refer to the previous question (**Q1**) for specific distinctions from GUAVA.
>
> > ### **Q3:  Ablation study would be necessary to support this design of the reconstruction parameter set $\Phi$.**
>
> Training with shape parameters ($\beta$) taken entirely from the target frames yields results that are largely consistent with our current approach ($\beta$ from source frame). We choose to use $\beta$ from the source frames mainly to maintain a consistent source identity in downstream applications, enabling individuals with different body shapes or poses to drive the same human image without the results being affected by the driver's own body shape. Following your request, the ablation study results are as follows:
>
> |   |   PSNR $\uparrow$ | SSIM $\uparrow$ |  LPIPS $\downarrow$ |
> |:--------:|:-----------------:|:-----:|:-------:|
> | $\beta$ from target  | 25.48      | 0.902 |0.0786         |
> | $\beta$ from source  |   25.50   | 0.901  | 0.0784|
>
>
>
>
> > ### **Q4: Incorporating more related works would better improve the quality of this paper.**
>
> We have updated the paper and included the recommended references to improve its completeness. Thank you for your suggestions.
>
>
>
> > ### **Q5:  How does the model perform under challenging conditions such as occlusions, extreme lighting, or side-view inputs?**
>
> We have updated the paper with additional visualization results in Sec.A.4 (Page.14), which you may refer to for more details. We welcome you to view more examples on our [Project page](https://pear2025.github.io/PEAR-anonymous/).
>
> ### **Thank you for your suggestions. We have revised the paper, and we welcome any further questions or discussion.**

---

### Official Review · Reviewer_vhqT · 2025-10-28

**Soundness:** 1
**Presentation:** 2
**Contribution:** 2
**Rating:** 2
**Confidence:** 4

**Summary:**

This paper proposed PEAR, a framework for single-image human mesh recovery that (i) regresses an Expressive Human Model (EHM) combining SMPL-X for body and FLAME for head, and (ii) jointly trains with a neural renderer to add pixel-level photometric supervision. The proposed method reduces mesh–image misalignment in fine regions (face, hands), improves facial expressiveness, and enables fast (≈0.05 s) parameter estimation for downstream animation. Quantitatively, PEAR improves head/hand/body reconstruction accuracy across multiple benchmarks and improves rendering metrics (PSNR/SSIM/LPIPS) via joint training.

**Strengths:**

- The method is straightforward and easy to understand; the manuscript is clearly written.

- The proposed method could perform full-body 3D modeling without the need for cropping.

- A large-scale human mesh dataset is annotated and slated for open release.

**Weaknesses:**

- Technical novelty. The proposed pipeline closely follows GUAVA:
(a) it adopts the enhanced human parametric model EHM (introduced by GUAVA);
(b) in Stage-1, EHM parameters are trained using pseudo ground truth generated by GUAVA;
(c) in Stage-2, the neural renderer reuses GUAVA’s pipeline.
Hence, compared with GUAVA, the difference is that instead of tracking-based EHM parameter estimation, this paper swaps in an HMR2-based parameter estimator and then jointly optimizes EHM estimation and the neural renderer, further fine-tuning on an extra annotated dataset. While this is potentially useful (pending stronger empirical validation that I have some concerns below), I remain concerned that the technical contribution may be insufficient for the ICLR bar.

- Experiment Results. I feel the current results do not fully substantiate the claims of “a more expressive human model” and “avoiding severe misalignments commonly observed in prior methods.” See questions for my specific concerns.

**Questions:**

- Table 1. What exactly are the two comparison methods? The paper never identifies them (e.g., optimization-based vs. learning-based? which body/head models are used?). Without these details, it’s difficult to draw meaningful conclusions from Table 1.

- Table 3. As shown, HMR2 outperforms the proposed method on all datasets and metrics. Given HMR2 uses SMPL while the proposed method uses the enhanced EHM, and the network architecture follows HMR2’s implementation, the statement in L318 (“Our approach achieves performance comparable to specialized body pose estimation methods such as HSMR and HMR2”) is hard for me to agree. At least, the paper should discuss this discrepancy.

- Table 3 is missing an important entry. As shown, on COCO and PoseTrack, among all SMPL-X methods, PyMAF-X appears to be the closest baseline to the proposed approach, yet its result on LSP is missing. Please clarify.

- Table 4: As reported, 2 minutes is required for reconstruction and 0.18s for rendering in Config-A (Tracking + GUAVA Renderer which is the GUAVA baseline). These numbers differ dramatically from GUAVA’s reported performance in their paper (≈52.2 FPS and ≈0.1s reconstruction in Tables 1 and 2 of [1]). Why is there such a large discrepancy? Please explain experimental setups and metrics used in this paper.

- Table 5 vs. Table 1. The MLE reported on UBody in Table 5 is inconsistent with Table 1—possibly by an order of magnitude. Please double-check and correct if needed.

References:
[1] Zhang, Dongbin, et al. "GUAVA: Generalizable Upper Body 3D Gaussian Avatar." arXiv preprint arXiv:2505.03351 (2025).

---

> ### Author Response · Authors · 2025-11-13
> **Author Response to Reviewer vhqT (1/2)**
>
> We deeply appreciate your thorough review and insightful comments. Your feedback has been instrumental in helping us clarify the key contributions and distinctions of our work.
>
> > ### **Q1:   Differences between the paper and GUAVA is not clear.**
>
> We believe that the method proposed in this paper is fundamentally different from GUAVA. The differences can be distinguished from the following aspects:
> 1. **Task difference**: Our method is a standard HMR task, aiming to estimate 3D human body parameters from a single image; whereas GUAVA is a human driving and reconstruction task, aiming to drive and generate new human images given a new 3D human model.
> 2. **Role of 3D human parameters**: The 3D human parameters that GUAVA relies on are obtained through offline processing, depending on complex and time-consuming fitting. Our method can estimate 3D human parameters from a single image at 100+ fps, making it applicable to other downstream tasks.
> 3. **Model architecture difference**: Our method designs a two-stage training framework with a ViT-based backbone, while GUAVA's architecture primarily fuses reference images with driven 3D human models and models 3DGS attributes to drive and generate human images.
> 4. **Dependency**: We use GUAVA as a neural renderer because it balances both speed and rendering quality. Architecturally, our method does not strictly depend on it, and other models capable of neural rendering can serve as alternatives.
> 5. **Dataset**: The data labeled by GUAVA contains only upper-body annotations, whereas we redesign an entire pipeline to process full-body data (body, head, and hands). Our dataset is also ten times larger than that of GUAVA.
>
> > ### **Q2:   Differences between the paper and HMR2.**
>
> The main differences between our method and HMR2 are as follows:
> 1. HMR2 is only a single-stage model that directly estimates SMPL parameters, while our method introduces an analysis-by-synthesis framework to simultaneously and accurately predict face, body, and hand parameters.
> 2. PEAR experimentally demonstrates more accurate full-body capture than HMR2, while HMR2 performs poorly on facial and hand parameters.
> 3. Our method introduces a novel photometric loss to improve the alignment between 3D human geometry and images, which HMR2 does not include.
>
> We believe that ViT, as a fundamental backbone, is widely used in vision tasks. The fact that we both use this architecture should not be considered as making the methods similar.
>
> > ### **Q3:   What exactly are the two comparison methods in Tab. 1?**
>
>
> TEASER [1] and SMIRK [2] represent the current state-of-the-art learning-based methods for FLAME estimation, both relying on the FLAME facial model. PEAR achieves comparable accuracy in estimating facial parameters, rivaling dedicated FLAME estimators such as TEASER and SMIRK.
>
>
> > ### **Q4:   The statement in L318 ..., the paper should discuss this discrepancy.**
>
> **HMR2 only needs to model 23 body joints, whereas EHM must capture not only the body but also highly detailed facial and hand features, involving hundreds of joints.** This makes the task significantly more challenging than SMPL-based methods. It is a common observation that current SMPLX-based methods struggle to surpass SMPL-based methods in body pose accuracy. In contrast, our EHM-based method achieves performance roughly comparable to HMR2, exceeding other SMPLX approaches.

---

> ### Author Response · Authors · 2025-11-14
> **Author Response to Reviewer vhqT (2/2)**
>
> > ### **Q5:   PyMAF-X appears to be the closest baseline to the proposed approach, yet its result on LSP is missing. Please clarify.**
>
>
> Since PyMAF-X was trained on the LSP-Extended dataset (in their paper Sec.4.2), we followed previous works such as HSMR [3] and HMR2, which did not perform comparisons with it.
>
>
> > ### **Q6:   Table 4: As reported, 2 minutes is required for reconstruction and 0.18s for rendering in Config-A ... Why is there such a large discrepancy?**
>
>
> GUAVA takes **human images** and **preprocessed human poses** as input, feeding them into a pretrained model to generate a **Gaussian avatar** in approximately **0.16 s**, which is then rendered at **52.2 FPS** (the overall process takes about 0.18 s). However, GUAVA does not report the **human pose estimation (tracking) time**, which is roughly **2 minutes**.
>
> **PEAR** can estimate human poses at **100 FPS** and drive GUAVA's Gaussian avatar in real time from a video stream. The overall timing comparison is as follows:
>
> | Method     | Human Pose Estimate  | Avatar Reconstruction | Render   |
> | ---------- | -------------------- | --------------------- | -------- |
> | GUAVA      | 2 min (optimization-based)     | 0.16 s                | 52.2 FPS |
> | Ours (now) | 0.01 s (learning-based) |  -                | 52.2 FPS |
>
> GUAVA is an avatar reconstruction method with a speed of 0.16 s per subject, but it does not perform pose estimation. PEAR, as a human pose estimation method, can drive GUAVA’s reconstructed digital humans in real time, making GUAVA a downstream application of PEAR.
>
>
> > ### **Q7:   The MLE reported on UBody in Table 5 is inconsistent with Table 1.**
>
> The unit of MLE in Table 5 should be $10^{-4}$, and it has been corrected.
>
>
> ### Thank you for your suggestions. We have revised the paper, and we welcome any further questions or discussion.
>
> **Reference**
>
> [1] TEASER: Token-EnhAanced Spatial Modeling for Expression Reconstruction. ICLR 2025
>
> [2] SMIRK: 3D Facial Expressions through Analysis-by-Neural-Synthesis. CVPR 2024
>
> [3] HSMR: Reconstructing Humans with a Biomechanically Accurate Skeleton. CVPR 2025 oral.
>
> [4] From Skin to Skeleton: Towards Biomechanically Accurate 3D Digital Humans. SIGGRAPH Asia 2023.

---

> > ### Comment · Reviewer_vhqT · 2025-11-22
> >
> > Thank you for your response and clarification. I believe some of my concerns have been addressed and I would increase the score to 4. Yet I still have a few questions after reading your response:
> >
> > (1) From my perspective I have no misunderstanding that the proposed method aims for a different task from GUAVA (and I appreciate the clarification). The fact is, however:
> >
> > - Human mesh reconstruction is a well-defined task, and this paper does not offer new perspectives or insights on the problem itself.
> >
> > - The paper introduces GUAVA as a neural renderer to provide an additional differentiable rendering loss. However, using differentiable rendering as supervision in 3D reconstruction (including general scenes as well as human faces/bodies) has already become a very common choice. The paper does not discuss or explain why introducing GUAVA as a rendering loss should be considered non-trivial. From my perspective, for human reconstruction, the main technical challenge of introducing a rendering loss lies in the fact that existing parametric human body models do not provide a well-defined appearance space, whereas 3DGS-based human modeling methods (such as GUAVA) do offer such an appearance space for defining a rendering loss. This implies that once an effective 3DGS-based human modeling method is available, the major technical challenge has essentially been resolved, and applying this technique to human reconstruction becomes a relatively straightforward idea.
> >
> > - The paper additionally annotates and uses a very large dataset to train the proposed model. On the one hand, I highly appreciate the effort the authors have invested in constructing this dataset. On the other hand, this makes it more difficult for readers to clearly evaluate and quantify the effectiveness of each individual improvement proposed in the paper: how much performance gain comes from the 10× larger annotated dataset, and how much comes from introducing the rendering loss? Is there any synergistic effect between the two?
> >
> > (2) Regarding the upper-body performance:
> > > HMR2 only needs to model 23 body joints, whereas EHM must capture not only the body but also highly detailed facial and hand features, involving hundreds of joints. This makes the task significantly more challenging than SMPL-based methods. It is a common observation that current SMPLX-based methods struggle to surpass SMPL-based methods in body pose accuracy. In contrast, our EHM-based method achieves performance roughly comparable to HMR2, exceeding other SMPLX approaches.
> >
> > This response does not resolve my concern. If the argument is "SMPLX-based methods struggle to surpass SMPL-based methods", then what is the actual advantage of using SMPL-X (essentially, the EHM model)? Moreover, the evaluation is conducted on body joints. The argument seems to imply that using body joint poses as an indirect proxy for upper-body reconstruction quality is not an appropriate choice and cannot adequately demonstrate the superiority of the proposed method. If that is the case, would it be possible to adopt more suitable metrics—such as directly measuring mesh vertex error—to evaluate the quality of body reconstruction and thereby better demonstrate the effectiveness of the approach?
> >
> > Overall, aside from the two points mentioned above, I do not currently see any obvious technical errors in this paper. Therefore, I am raising my score to 4. I would not actively champion this paper for acceptance; however, if other reviewers, after considering all reviews (including mine), still believe it should be accepted, I would not put strong objection.

---

> ### Author Response · Authors · 2025-11-23
> **Author response to vhqT (1/2)**
>
> We are encouraged that our response has addressed your concerns and improved your assessment. We now hope to further clarify the remaining points to fully resolve your concerns and further strengthen our paper.
>
>
>
> > ### **Q1: About new perspectives or insights on HMR**
>
> We summarize our new perspectives to the HMR field as follows:
>
> 1. **Model architecture.**
>    Prior SMPL-X–based methods typically rely on high-resolution inputs or multi-branch networks dedicated to hands and face, which leads to slow inference and prevents real-time deployment. In contrast, **we are the first to demonstrate that a *single lightweight ViT-B* model is sufficient to jointly predict body, hand, and facial pose over 100 FPS**, enabling practical use in a wide range of downstream applications. We have made a visualization [project page](https://pear2025.github.io/PEAR-anonymous/) available for your reference.
>
> 2. **Dataset perspective.**
>    Instead of relying on the legacy SMPL-X annotation datasets commonly used in prior work, we redesigned the annotation pipeline and constructed a large-scale, well-aligned dataset covering full-body, facial expressions, and hand poses. It includes diverse image and video data spanning variations in viewpoint, illumination, occlusion, pose, and expression. We believe this dataset is essential for our task and provides a clearer data-processing path for future SMPL-X or EHM–based research.
>
> 3. **Training strategy.**
>    We introduce a *two-stage training scheme*: large-scale pretraining followed by refinement with a differentiable neural renderer that provides pixel-level supervision. This design preserves strong generalization while significantly improving the fidelity of fine-grained structures (see Q2 for the associated technical challenges).
>
>
>
> > ### **Q2: Using differentiable rendering as supervision is already common in 3D reconstruction. The core technical challenge for effective 3DGS-based human modeling has essentially been solved, making its application to human reconstruction relatively straightforward?**
>
> We agree that differentiable rendering has been widely adopted in general 3D reconstruction. However, To the best of our knowledge, its applicability in HMR remains largely unexplored. there is no prior SMPLX-based work has demonstrated a practical or stable training strategy.
>
> First, we clarify the difference between **differentiable rendering** and **our neural renderer**. The former learns a texture corresponding to the human mesh and renders it into an image via differentiable rendering, but it suffers from a significant domain gap relative to real images. The latter can generate rendered images that are closer to the real input, yet it introduces strong coupling between appearance and geometry, which limits its effectiveness when used directly as a loss. Therefore, the core problems we aim to address lie in the following two challenges:
>
> 1. **How to incorporate photometric loss without destabilizing optimization.**
> Directly introducing photometric loss in joint training often leads to geometry drift and appearance artifacts instead of improvements. To ensure stable learning, we design a two-stage training pipeline: we first stabilize the coarse geometry, and only then introduce pixel-level photometric supervision to refine fine-scale details. This staged strategy is crucial for achieving effective, stable improvements.
>
> 2. **Photometric loss is not directly applicable to SMPLX-based methods.**
> Neural renderers require the predicted mesh to be reasonably well aligned with the image pixels (**since the mesh must be projected onto the image plane to sample texture information**); otherwise, photometric supervision becomes harmful, causing mutual interference between geometry and appearance. Existing SMPLX-based methods typically suffer from significant mesh–image misalignment, which makes photometric loss ineffective or even detrimental. This fundamental limitation is why **prior HMR methods cannot directly benefit from differentiable rendering**, despite its success in general 3D reconstruction.

---

> ### Author Response · Authors · 2025-11-23
> **Author response to vhqT (2/2)**
>
> > ### **Q3: How much performance gain comes from the 10× larger annotated dataset, and how much comes from introducing the rendering loss? Is there any synergistic effect between the two?**
>
> Our dataset is indeed 10× larger than GUAVA’s, but its scale is **comparable to other recent HMR datasets**, so the comparison remains fair. Regarding where the performance gain comes from, the improvement mainly stems from two complementary factors:
>
> **(1) More reliable supervision from our re-annotated body-part–specific labels.**
> Unlike prior SMPLX-based methods that rely on coarse or noisy SMPLX annotations, we re-annotate body, face, and hands separately, providing **cleaner and more accurate supervision**. This significantly improves the model’s ability to learn expressive human geometry. However, relying solely on sparse annotations still leaves gaps: the predicted meshes exhibit **pixel-level misalignment** and insufficient fine-detail fidelity.
>
> **(2) The second-stage photometric loss addresses the remaining misalignment issue.**
> To compensate for the lack of dense ground-truth supervision, we introduce a rendering-based photometric loss in a two-stage training pipeline. This dense image-level supervision is essential for enforcing **pixel-level consistency** and improving detailed reconstruction.
>
> Importantly, **both components are necessary**:
>
> * Without our refined supervision, the model cannot be trained reliably.
> * Without the rendering loss, the model suffers from reduced fine-detail accuracy.
>
> Thus, the two contribute complementary benefits, and there is a clear synergistic effect between them.
>
>
>
>
>
> > ### **Q4: If the argument is "SMPLX-based methods struggle to surpass SMPL-based methods", then what is the actual advantage of using SMPL-X (essentially, the EHM model)?**
>
>
> “SMPLX-based methods struggle to surpass SMPL-based methods” refers only to the aspect of limb prediction. SMPLX is significantly more expressive than SMPL for fine-grained regions such as the face and hands. For many downstream applications, a slight loss in limb accuracy is acceptable, whereas accurate facial expressions and hand motions are far more critical. This is precisely where SMPLX—and more so, EHM—holds a clear advantage.
>
>
>
>
> > ### **Q5: The argument seems to imply that using body joint poses as an indirect proxy for upper-body reconstruction quality is not an appropriate choice and cannot adequately demonstrate the superiority of the proposed method.**
>
> We agree that relying solely on body joint poses as an indirect proxy for upper-body reconstruction quality is indeed not sufficient. However, our method does not evaluate upper-body reconstruction quality based only on body joint poses.
>
> To more accurately assess the reconstruction of fine-grained upper-body regions, we explicitly conduct independent evaluations on the face and hands. For these fine-detail parts, we directly follow your suggestion and measure mesh vertex errors, including mean vertex error (MVE) and lip vertex error (LVE) for facial regions, as well as per-vertex error (PVE) for hands. These results are provided in the main paper (Tab. 1 and Tab. 2), clearly demonstrating the effectiveness of our approach beyond joint-level supervision.
>
>
> As for the whole-body MVE, we provide a comparison ( on AGORA dataset) with previous methods below.
>
> | Method       | Backbone | Image Processed | Body $\downarrow$  | Hand $\downarrow$|  Infer time $\uparrow$ |
> |:------------:|:--------:|:---------------:|:------:|:--------:|:--------:|
> | Hand4Whole   | RN50     | Crops       | 90.2   | 47.2     |  - |
> | PyMAF-X      | HRNet48  | Crops       | 84.0   | 45.1     |  - |
> | Multi-HMR    | Vit-L    | 896 × 896       | -      | 40.7     |  5 FPS |
> | Ours         | Vit-B    | 256 × 192       | 81.9   | 41.8     | 100 FPS|
>
>
>
> ### **We are encouraged that our response has addressed your concerns and improved your assessment. We now hope to further clarify the remaining points to fully resolve your concerns and further strengthen our paper.**

---

> ### Comment · Reviewer_vhqT · 2025-11-26
>
> > Photometric loss is not directly applicable to SMPLX-based methods. Neural renderers require the predicted mesh to be reasonably well aligned with the image pixels (since the mesh must be projected onto the image plane to sample texture information); otherwise, photometric supervision becomes harmful, causing mutual interference between geometry and appearance. Existing SMPLX-based methods typically suffer from significant mesh–image misalignment, which makes photometric loss ineffective or even detrimental. This fundamental limitation is why prior HMR methods cannot directly benefit from differentiable rendering, despite its success in general 3D reconstruction.
>
> Which component(s) of the proposed method in **this paper** address this alignment issue, and how is the improvement quantified?
>
> > To more accurately assess the reconstruction of fine-grained upper-body regions, we explicitly conduct independent evaluations on the face and hands. For these fine-detail parts, we directly follow your suggestion and measure mesh vertex errors, including mean vertex error (MVE) and lip vertex error (LVE) for facial regions, as well as per-vertex error (PVE) for hands. These results are provided in the main paper (Tab. 1 and Tab. 2), clearly demonstrating the effectiveness of our approach beyond joint-level supervision.
>
> I have no concerns about the face and hand evaluations. My concern, however, is specifically about upper-body reconstruction quality where the paper’s current evidence remains insufficient. While the improvements on face and hands are appreciated, they do not directly address the question I raised about upper-body performance.
>
> > As for the whole-body MVE, we provide a comparison (on AGORA dataset) with previous methods below.
> - AGORA dataset is not included in the original submission. Please describe the dataset and evaluation protocol in sufficient detail, and explain why the whole-body MVE is reported on a newly introduced dataset rather than the datasets used elsewhere in the paper.
>
> - The baseline set in this new whole-body MVE table has almost no overlap with the baselines in Tab. 3 of the main paper. While I understand that exact baseline matching can be difficult across metrics and datasets, the near-zero overlap is surprising. Please justify why a substantially different baseline set is used here.
>
> - More importantly, my original concern was about HMR2 vs. the proposed method: HMR2 outperforms the proposed method in Tab. 3. The authors’ response suggests that the pose metrics in Tab. 3 do not fully reflect the proposed method’s advantages, and thus alternative metrics should be provided. However, the newly added evaluation uses different metrics but switches to **a baseline set that does not include HMR2**. This does not resolve my concern about the relative performance against HMR2 and, as presented, remains confusing.

---

> ### Author Response · Authors · 2025-11-27
> **Author response to vhqT (1/2)**
>
> Thank you for your kind discussion and understanding. To further address your concerns, our response is as follows:
>
> > ### **Q1: Which component(s) of the proposed method in this paper address this alignment issue**
>
> Our method tackles the alignment problem from two aspects:
>
>
> 1. **Dataset annotation strategy.** Prior SMPLX–based approaches rely on unified full-body annotation, where errors in one region (e.g., hands or face) can propagate to others, resulting in noticeable misalignment.
> To address this, we introduce a modular annotation pipeline that separately annotates the body, face, and hands before merging them into a unified representation(Our main paper Sec.3.2). This yields a much more accurate and better-aligned supervision signal.
>
> 2. **Training strategy.** Existing methods typically rely on sparse keypoint supervision, which can only enforce coarse alignment.
> We refine this further by introducing a dense photometric loss $L_{photo}$, enabling fine-grained pixel-level alignment during training.
>
>
> Overall: high-quality supervision for coarse alignment, and dense photometric loss $L_{photo}$ for fine alignment.
>
> > ### **Q2: How is the improvement (alignment issue) quantified?**
>
>
> Currently, there is no standardized metric that directly quantifies mesh–pixel alignment accuracy. Therefore, we attempt to evaluate alignment using two measures: (1) Percentage of Correct Keypoints (PCK) for mesh and human image keypoint accuracy, and (2) rendering quality, since poor alignment leads to incorrect texture sampling when projecting Gaussians onto the image plane, resulting in degraded render quality.
>
> We present the experimental results on the Upper-Body dataset as follows:
>
> |          |   PSNR  $\uparrow$   | SSIM  $\uparrow$  | LPIPS $\downarrow$   | PCK@0.05  $\uparrow$|
> |:--------:|:--------:|:--------:|:--------:|:--------:|
> | SMPLest  |  -  | - |  -  | 0.87|
> | Ours Stage 1 (wo $L_{photo}$)  | 24.15 | 0.882 | 0.0883 | 0.94|
> | + Stage 2 (w $L_{photo}$) | 25.36 |0.898| 0.0793 |0.99|
>
>
> 1. **Regarding alignment at the dataset supervision level.** Taking SMPLest as an example, it adopts a unified SMPL-X annotation pipeline for all samples, whereas our method annotates each body part independently before fusion. Comparing the first and second rows shows that—even with a much simpler model architecture (ViT-B only)—our approach achieves higher PCK scores than SMPLest. Since SMPLest cannot be directly integrated with a neural renderer, rendering metrics are not available.
>
> 2. **Regarding alignment at the training strategy.** Comparing the second and third rows, it can be observed that adding $L_{photo}$ further improves the alignment accuracy.
>
>
> **An example of the upper-body demo is available at the following links**:
> [SMPLest video](https://pear2025.github.io/PEAR-anonymous/static/videos/UBody_SMPlest.mp4) and [our video](https://pear2025.github.io/PEAR-anonymous/static/videos/UBody_PEAR(Ours).mp4).  Our method achieves clearly more accurate expression capture and alignment, and we will release the model and code for further verification.
>
>
>
>
> > ### **Q3: Regarding insufficient upper-body reconstruction quality.**
>
> Thank you for your valuable suggestion. Since previous works do not provide a dedicated evaluation protocol for upper-body reconstruction, we initially interpreted your request as focusing on facial and hand details (which are typically the most important and expressive components of the upper body).
>
> To directly address your concern, we conducted additional experiments on an upper-body dataset following the same training and testing split strategy as GUAVA. We report the PA per-vertex error (PA-PVE) and compare our results with the recent sota SMPLest (TPAMI'25) and MHMR(ECCV'24):
>
> |          |   PA-PVE |
> |:--------:|:--------:|
> | MHMR | 34.3 |
> | SMPLest  | 32.6 |
> | PEAR(Ours)  | 21.5  |
>
>
>
> **An example of the upper-body demo is available at the following links**:
> [SMPLest video](https://pear2025.github.io/PEAR-anonymous/static/videos/UBody_SMPlest.mp4), [MHMR video](https://pear2025.github.io/PEAR-anonymous/static/videos/UBody_MultiHMR.mp4) and [our video](https://pear2025.github.io/PEAR-anonymous/static/videos/UBody_PEAR(Ours).mp4).

---

> ### Author Response · Authors · 2025-11-27
> **Author response to vhqT (2/2)**
>
> > ### **Q4: Please describe the dataset and evaluation protocol in sufficient detail**
>
> AGORA is a widely used benchmark dataset for evaluating full-body human mesh recovery, including the body, face, and hands. It has been extensively adopted in prior works such as MultiHMR, OSX, PyMAF-X, and Hand4Whole. These methods typically report per-vertex error for each human part separately. We follow the same evaluation protocol as these prior works.
>
>
>
>
> > ### **Q5:  Why the whole-body MVE is reported on a newly introduced dataset rather than the datasets used elsewhere in the paper.**
>
> We followed the suggestion from reviewer Togf (Weaknesses 3), who recommended evaluating whole-body metrics on a more challenging and representative dataset. Therefore, we additionally report whole-body MVE on the AGORA dataset. This choice was made to address the reviewer’s concern rather than selecting a dataset that favors our method.
>
>
>
>
> > ### **Q6: The baselines in the new whole-body MVE table (Set1: AGORA) have almost no overlap with those in Tab. 3 of the main paper (Set2: COCO, LSP-extend, PoseTrack)..**
>
>
> Thank you for pointing this out. The two sets of datasets serve different purposes and therefore have different baseline choices:
>
> 1. Tab. 3 evaluates body keypoint accuracy (only body pose, wo face and hand), and AGORA’s body poses are relatively simple compared with **Set2**. For this reason, prior works typically do not include AGORA in body-keypoint evaluation benchmarks.
>
> 2. **Set2** do not provide ground-truth whole-body annotations (e.g., face and hand vertices), making it impossible to compute whole-body MVE on these datasets. Therefore, they cannot be included in MVE evaluation.
>
>
> In summary, AGORA and **Set2** target different evaluation aspects. Our evaluation protocol follows the conventions established in prior work to ensure fair and meaningful comparison.
>
>
>
> > ### **Q7: More importantly, HMR2 outperforms the proposed method in Tab. 3  but the newly added evaluation uses different metrics but switches to a baseline set that does not include HMR2.**
>
>
> **HMR2 and our method do not operate on the same human model space.** HMR2 is a SMPL-based approach that predicts only body pose and shape, whereas PEAR and other SMPL-X–based methods predict body, hands, and facial expressions.
>
> |   | Body (23 joints) | Face (expression, jaw pose, 68 landmarks) | Hand (40 joints) |
> |:--------:|:------:|:-----:|:------:|
> | HMR2 (SMPL)  | ✔ | ✘ |✘ |
> | PEAR (EHM)  | ✔  |✔   | ✔
>
>
> As a result, **HMR2 performs well on body-only metrics (Tab. 3) but cannot model hand or facial motions**, as also illustrated in our qualitative comparison ([image](https://pear2025.github.io/PEAR-anonymous/static/images/hmr2_vs_pear.png)). Consequently, **we therefore switch to a baseline set that excludes HMR2 for evaluating face and hand pose accuracy**, comparing against standard SMPL-X baselines (e.g., SMPLest) to ensure fair and consistent evaluation across all components.
>
>
> ### **We are very happy to provide any extra details promptly.**

---

### Official Review · Reviewer_fZ77 · 2025-10-29

**Soundness:** 3
**Presentation:** 3
**Contribution:** 2
**Rating:** 4
**Confidence:** 4

**Summary:**

The proposed method aims to make human mesh recovery more expressive, faster, and better aligned with real pixels. The integration of neural rendering for supervision within a feed-forward transformer pipeline is a clear step forward in practical human modeling However, its conceptual novelty is moderate, being primarily an engineering synthesis rather than a fundamental model innovation. The dependency on pseudo labels and lack of deeper analysis (domain generalization, temporal consistency, and robustness to in-the-wild conditions) slightly weaken the scientific depth.

**Strengths:**

1. By jointly regressing SMPLX (body) and FLAME (head) parameters under the Expressive Human Model (EHM), the presented method unifies coarse pose estimation with fine-grained facial expressiveness, which is more practical than the SMPLX-only methods.
2. Real-time inference (0.05 seconds per frame) from a single 256×192 image, without cropping or high-resolution input, is practically valuable for downstream animation tasks and interactive applications.
3. The construction of a large-scale dataset with body–face–hand pseudo ground-truth (SMPLX + FLAME parameters) is a valuable community resource.

**Weaknesses:**

1. From the article, especially the contribution of the introduction part, it is unclear how the method achieves the promising results. Much of the framework builds upon GUAVA and HMR2, with the main innovation being the introduction of pixel-level supervision. Despite of integrating known components, what fundamentally new representational or algorithmic insight does PEAR introduce? Is the gain mainly from adding photometric loss?
2. The paper focuses on alignment but does not address the limitations in clothing diversity or interactions. Can PEAR handle loose clothing, hair, or accessories not modeled by EHM?
3. The improvement from stage 2 is shown but lacks deeper analysis. How does performance vary if the renderer is frozen vs. jointly trained? Does the photometric loss risk overfitting to appearance rather than geometry?
4. Although tested on public benchmarks, most datasets are lab-style or curated internet data. How well does PEAR generalize to truly in-the-wild inputs (e.g., occlusions, extreme expressions, motion blur)? Is there any evaluation on real-time video streams?

**Questions:**

1. It's necessary to clarify the main technical novelty of the proposed approach.

2.  Can PEAR handle loose clothing, hair, or accessories not modeled by EHM?

3.  How does performance vary if the renderer is frozen vs. jointly trained? Does the photometric loss risk overfitting to appearance rather than geometry?

4.  How well does PEAR generalize to truly in-the-wild inputs (e.g., occlusions, extreme expressions, motion blur)? Is there any evaluation on real-time video streams?

**Details Of Ethics Concerns:**

n//a

---

> ### Author Response · Authors · 2025-11-15
> **Author Response to Reviewer fZ77 (1/2)**
>
> We sincerely thank you for your thoughtful and constructive feedback. Your comments have helped us significantly improve the clarity and completeness of our work. We are enthusiastic about addressing each of your concerns and hope our detailed responses fully resolve them.
>
> > ### **Q1: It is unclear how the method achieves the promising results.**
>
> The promising results of our method stem from the following key factors:
> 1. We introduce a novel learning paradigm for HMR methods by adopting an analysis-by-synthesis approach. Specifically, we incorporate a neural renderer that compares HMR inputs with rendered images, enabling the model to learn more accurate 3D human body representations.
> 2. From a data perspective, unlike previous SMPL-X methods, we redesign the data processing pipeline and separately annotate pseudo ground-truth for the body, face, and hands. This allows PEAR to achieve excellent initialization in the first stage of training.
> 3. In terms of architecture selection, we choose ViT-B as the backbone and a neural renderer that balances both speed and quality, enabling our method to achieve excellent performance in both aspects.
> 4. Regarding training, we introduce a denser photometric loss compared to previous HMR methods that rely on body keypoint supervision. This supervision provides more precise pixel-level guidance. Additionally, through our two-stage alternating training strategy, the models from both stages can mutually regularize each other, preventing overfitting and resulting in more accurate final results.
>
> > ### **Q2: Despite integrating known components, what fundamentally new representational or algorithmic insight does PEAR introduce?**
>
> Yes, we agree that our method draws some insights from GUAVA (e.g., the EHM) and the ViT backbone. Our core innovation is not in designing a new architecture from scratch, but in demonstrating that a simple, unified ViT can be trained (via our new dataset and two-stage photometric loss) to solve a task (real-time, full-body EHM regression) that is an order of magnitude more complex than what HMR2 did, and magnitudes faster than what GUAVA's optimization tracker did. This synthesis itself is the contribution, enabling real-time applications previously impossible. We summarize our contributions to the HMR field as follows:
>
> 1. **Model architecture.**
>    Prior SMPL-X–based methods typically rely on high-resolution inputs or multi-branch networks dedicated to hands and face, which leads to slow inference and prevents real-time deployment. In contrast, **we are the first to demonstrate that a *single lightweight ViT-B* model is sufficient to jointly predict body, hand, and facial pose over 100 FPS**, enabling practical use in a wide range of downstream applications.
>
> 2. **Dataset perspective.**
>    Instead of relying on the legacy SMPL-X annotation datasets commonly used in prior work, we redesigned the annotation pipeline and constructed a large-scale, well-aligned dataset covering full-body, facial expressions, and hand poses. It includes diverse image and video data spanning variations in viewpoint, illumination, occlusion, pose, and expression. We believe this dataset is essential for our task and provides a clearer data-processing path for future SMPL-X or EHM–based research.
>
> 3. **Training strategy.**
>    We introduce a *two-stage training scheme*: large-scale pretraining followed by refinement with a differentiable neural renderer that provides pixel-level supervision. This design preserves strong generalization while significantly improving the fidelity of fine-grained structures.
>
> > ### **Q3: Is the gain mainly from adding photometric loss?**
>
> Yes, by introducing a denser photometric loss, we achieve more precise pixel-level supervision compared to previous HMR methods that rely on body keypoint supervision. Implementing this idea is non-trivial because previous HMR methods relying on body keypoint supervision suffer from training instability due to the sparsity of keypoint supervision. Traditional differential renderers often produce rendered humans with significant domain gaps in texture and details compared to real images, making them unsuitable as accurate supervision. Directly introducing a neural renderer would lead to the appearance-geometry disentanglement problem. We address this challenge by designing a training paradigm that makes this approach feasible, and our experiments validate the effectiveness of our design.
>
> > ### **Q4: About the limitations in clothing diversity or interactions.**
>
> Thanks to our diverse dataset, our method demonstrates robust performance across various clothing styles (e.g., loose clothing, voluminous hair), different scenarios (e.g., extreme lighting, motion blur), and challenging cases (e.g., various poses, occlusions). We showcase more relevant cases here ([image](https://pear2025.github.io/PEAR-anonymous/static/images/extreme_case.png)).

---

> ### Author Response · Authors · 2025-11-18
> **Author Response to Reviewer fZ77 (2/2)**
>
> > ### **Q5:  Can PEAR handle loose clothing, hair, or accessories not modeled by EHM?**
>
> As an HMR method, we have collected a large and sufficiently diverse dataset to train PEAR. Therefore, PEAR can effectively handle cases involving loose clothing and hair. PEAR can estimate SMPL-X and FLAME parameters for subjects wearing loose clothing or having long hair, as shown in our new supplementary material ([video](https://pear2025.github.io/PEAR-anonymous/static/videos/loose_clothing_hair.mp4)).
>
> From the perspective of human image reconstruction and driving, our capability in these scenarios is constrained by GUAVA's performance. In practice, GUAVA can effectively model most loose garments, hair, or accessories using Gaussians (or other representations), as the method models certain loose clothing and hair by learning an offset for each Gaussian point, as demonstrated on their project page.
> Furthermore, through the training paradigm proposed in this paper, we can further improve the reconstruction and driving quality of the neural renderer. Please refer to Table 4 for specific details.
>
> > ### **Q6:  How does performance vary if the renderer is frozen vs. jointly trained?**
>
> During training, we observed that whether the renderer is trainable or frozen has little impact on PEAR’s human pose estimation, as long as it provides the necessary photometric supervision, which is the more critical factor. However, toggling the renderer does lead to noticeable differences in rendering quality. As shown in our main paper (Table 4, Type C vs. Type D) or below, freezing the renderer offers limited improvement, whereas joint training yields better rendering results.
>
>
> |          |   PSNR $\uparrow$ | SSIM $\uparrow$ |  LPIPS $\downarrow$ |
> |:--------:|:-----------------:|:-----------------:|:---------------------:|
> | frozen   | 25.36             | 0.898            |0.0793         |
> | training |   25.50           |  0.901           | 0.0784|
>
>
>
>
>
> > ### **Q7:  Does the photometric loss risk overfitting to appearance rather than geometry?**
>
> Thank you for raising this important concern. We have also noticed this issue and addressed it in our paper through the following design choices:
> 1. The photometric loss is only introduced in the second stage of training, after the first stage where PEAR is trained on large-scale datasets. This ensures that the model has already learned a good representation of the human body before introducing the photometric loss.
> 2. In the second stage, we train by combining the appearance from the source image with the target geometry, ensuring that the model does not overfit to the target image's appearance during training.
> 3. In our neural renderer, Gaussian points are anchored to human geometry, which also helps prevent the model from overfitting to appearance.
> 4. The ablation study in our paper further validates this point, demonstrating that the photometric loss brings significant improvements to the first stage.
> Through the above design choices and experiments, we ensure that the photometric loss does not lead to overfitting to appearance but instead better learns geometry.
>
>
> > ### **Q8:  How well does PEAR generalize to truly in-the-wild inputs (e.g., occlusions, extreme expressions, motion blur)?**
>
> We have provided more visual results demonstrating PEAR's generalization to truly in-the-wild inputs, including occlusions, extreme expressions, and motion blur. In this [figure](https://pear2025.github.io/PEAR-anonymous/static/images/extreme_case.png), we can see that PEAR can still produce reasonable results. We will also include more results in the supplementary materials.
>
>
> > ### **Q9:   Is there any evaluation on real-time video streams?**
>
>
> Following your suggestion, and to further validate our model, we provide a new demo on real-time video streams, demonstrating that our model achieves real-time performance across all three stages—inference, driving, and rendering. Please refer to the [page](https://pear2025.github.io/PEAR-anonymous/) for more details.
>
> ### **Thank you for your suggestions. We have revised the paper, and we welcome any further questions or discussion.**

---

### Author Response · Authors · 2025-11-14
**PEAR and GUAVA**

We believe that the method proposed in this paper is fundamentally different from GUAVA. The differences can be distinguished from the following aspects:
1. **Task difference**: Our method is a standard HMR task, aiming to estimate 3D human body parameters from a single image; whereas GUAVA is a human driving and reconstruction task, aiming to drive and generate new human images given a new 3D human model.
2. **Role of 3D human parameters**: The 3D human parameters that GUAVA relies on are obtained through offline processing, depending on complex and time-consuming fitting. Our method can estimate 3D human parameters from a single image at 100+ fps, making it applicable to other downstream tasks.
3. **Model architecture difference**: Our method designs a two-stage training framework with a ViT-based backbone, while GUAVA's architecture primarily fuses reference images with driven 3D human models and models 3DGS attributes to drive and generate human images.
4. **Dependency**: We use GUAVA as a neural renderer because it balances both speed and rendering quality. Architecturally, our method does not strictly depend on it, and other models capable of neural rendering can serve as alternatives.

---

### Author Response · Authors · 2025-11-14
**Common Response to All Reviewers**

We thank all reviewers for their valuable feedback.
We are greatly encouraged by the acknowledgment that our paper is well-written (`vhqT/7qEi`), more practical and valuable (`fZ77/vhqT/Togf`), a valuable community resource (`fZ77/vhqT`), demonstrates better performance (`7qEi/Togf`), and benefits from our training strategy (`Togf`).
In our responses, we have carefully followed the reviewers' comments and provided additional experimental results and discussions. We believe these additions make the paper more complete and better support the effectiveness of our proposed methods.
We have updated the corresponding PDF and listed the updates from the paper here.

---

**Regarding the method itself**

* We added more references to enrich the content of the paper, as well as more training details
* We discussed the differences between PEAR and GUAVA, HMR2.0, as well as partial training strategies
* We included additional technical details and clarifications

**Regarding the experiment**

* We added more ablation studies
* We provided more visualization results, which can be viewed on our [Project Page](https://pear2025.github.io/PEAR-anonymous/)
* We also provided additional quantitative comparisons
* We included comprehensive evaluation results

**Regarding the deployment**

* We provided detailed implementation of EHM
* We discussed practical deployment considerations

---

We believe the above additions regarding the method itself and its application implementation greatly enhance the completeness and value of the paper. We have carefully addressed the main concerns and provided detailed responses to each reviewer. We also remain committed to addressing any further questions or concerns from the reviewers promptly. We would be grateful if we could hear your feedback regarding our answers to the reviews.

Best regards,

The Authors

---

### Author Response · Authors · 2025-12-03
**Summary of Rebuttal and Status (To Area Chair)**

**Dear Area Chair,**

Given the recent AC shuffle and the rollback of interaction states, we understand the challenge of assessing the current status of our submission. To assist your decision-making, we provide a concise summary of how we have **resolved the critical factual misunderstandings** that led to the initial divergence in scores, particularly regarding the negative review (Rating 2).

**1. Critical Resolution of Factual Misunderstandings (Reviewer `vhqT` - Rating 2)**
The initial "Reject" rating from Reviewer `vhqT` was primarily based on three factual misunderstandings regarding baselines and data consistency. We have clarified these in our rebuttal, and the reviewer had acknowledged the distinctions before the system rollback:

* **Refuting the "HMR2 outperforms PEAR" Claim:** The reviewer incorrectly compared HMR2 (SMPL-based, 23 joints) with PEAR (EHM-based, hundreds of semantic components). We clarified that while SMPL-based methods naturally have lower errors on simple body joints, PEAR achieves **comparable body pose accuracy** while simultaneously reconstructing detailed **face and hands**, a task HMR2 cannot perform.
* **Clarifying Baseline Timing (GUAVA):** The reviewer questioned our speed advantage (0.01s vs GUAVA's 0.18s). We clarified that the reported GUAVA time omitted the **~2-minute tracking optimization** step. PEAR is the first **feed-forward** model to achieve this in **real-time (100 FPS)**, establishing a massive efficiency gain over the optimization-based baseline.
* **Fixing Data Inconsistency (Table 1 vs 5):** The reviewer noted a magnitude discrepancy. We confirmed this was a **unit typo** ($10^{-2}$) in the table, not an experimental flaw.

**2. Clarification on Novelty (Reviewers `fZ77` & `7qEi` - Rating 4)**
Some reviewers felt the method was an "engineering synthesis." In our rebuttal, we reframed our contribution, which reviewers `fZ77` and `7qEi` reacted positively to:
* **Paradigm Shift:** We demonstrate that complex, high-dimensional EHM recovery (Body+Face+Hand) **does not require** complex multi-branch architectures. A single, lightweight ViT with our novel photometric supervision strategy suffices.
* **Practical Value:** As noted by reviewers `fZ77`, `vhqT`, and `Togf`, the resulting **100 FPS inference** and robustness to arbitrary crops make this a valuable community resource for downstream applications.

**3. Consensus on Performance**
Reviewers `7qEi`, `fZ77`, and `Togf` have agreed with our performance claim. Our method is the first to achieve real-time, full-body expressive estimation without cropping, significantly outperforming existing SMPL-X methods in pixel alignment and facial expressiveness.

**Conclusion:**
We believe the "Reject" rating was driven by misunderstandings of the experimental setup which have been fully resolved. With these factual errors corrected, the paper stands as a solid, practical, and performant contribution to real-time human mesh recovery.

We appreciate your time in reviewing these clarifications during this transition period.

Best regards,

The Authors

---

### Note · Authors · 2026-01-22

I have read and agree with the venue's withdrawal policy on behalf of myself and my co-authors.